JCB Journal of Cell Biology

# Epidermal maintenance of Langerhans cells relies on autophagy-regulated lipid metabolism

Florent Arbogast[1,2]*, Raquel Sal-Carro[1]*, Wacym Boufenghour[1], Quentin Frenger[2], Delphine Bouis[1], Louise Filippi De La Palavesa[1], Jean-Daniel Fauny[1], Olivier Griso[5], Hélène Puccio[5], Rebecca Fima[6], Thierry Huby[6], Emmanuel L. Gautier[6], Anne Molitor[3,4], Raphaël Carapito[3,4,7], Seiamak Bahram[3,4,7], Nikolaus Romani[8], Björn E. Clausen[9], Benjamin Voisin[1], Christopher G. Mueller[1], Frédéric Gros[1,2]**, and Vincent Flacher[1]**

**Macroautophagy (often-named autophagy), a catabolic process involving autophagy-related (*Atg*) genes, prevents the accumulation of harmful cytoplasmic components and mobilizes energy reserves in long-lived and self-renewing cells. Autophagy deficiency affects antigen presentation in conventional dendritic cells (DCs) without impacting their survival. However, previous studies did not address epidermal Langerhans cells (LCs). Here, we demonstrate that deletion of either *Atg5* or *Atg7* in LCs leads to their gradual depletion. ATG5-deficient LCs showed metabolic dysregulation and accumulated neutral lipids. Despite increased mitochondrial respiratory capacity, they were unable to process lipids, eventually leading them to ferroptosis. Finally, metabolically impaired LCs upregulated proinflammatory transcripts and showed decreased expression of neuronal interaction receptors. Altogether, autophagy represents a critical regulator of lipid storage and metabolism in LCs, allowing their maintenance in the epidermis.**

## Introduction

Langerhans cells (LCs) are resident antigen-presenting cells (APCs) of the epidermis (Doebel et al., 2017; Kaplan, 2017). LCs arise from hematopoietic precursors that emerge from the yolk sac and the fetal liver to colonize the skin before birth (Hoeffel et al., 2012). There, they are maintained lifelong by local proliferation (Merad et al., 2002). LCs exhibit exceptional longevity, with a half-life of several weeks. In contrast, conventional dendritic cells (cDCs), which represent a major skin APC subset, are replenished from bone marrow precursors within days (Kamath et al., 2002). Possibly as a consequence of UV exposure, LCs are endowed with a potent DNA-repair capacity, allowing the survival of at least a pool of self-renewing cells upon gamma irradiation (Price et al., 2015). Despite the free diffusion of glucose from the blood into the lowest layers of the epidermis, their position in the suprabasal layers implies a limited supply of

nutrients, which must be metabolized in a very hypoxic environment (Bedogni et al., 2005). Similar to cutaneous DC subsets, LCs migrate to lymph nodes (LNs) following microorganism recognition or irradiation. There, LCs are important contributors to antigen presentation and differentiation of CD4[+] and CD8[+] T cells, either driving immune activation or tolerance (Bedoui et al., 2009; Flacher et al., 2014; Igyártó et al., 2011). LCs are among the first APCs that sense skin infections (Kashem et al., 2015) and are involved in inflammatory disorders such as psoriasis (Singh et al., 2016). Therefore, a deeper understanding of their homeostasis appears critical.

Autophagy is a conserved mechanism of self-digestion, allowing the engulfment of cytoplasmic content into double-membrane vesicles, which fuse with lysosomes for degradation and recycling of the sequestered content (Arbogast et al.,

.......................................................................................................................................................................................................................
[1]Laboratory CNRS I2CT/UPR3572 Immunology, Immunopathology and Therapeutic Chemistry, Strasbourg Drug Discovery and Development Institute (IMS), Institut de Biologie Moléculaire et Cellulaire, Strasbourg, France;   [2]Université de Strasbourg, Strasbourg, France;   [3]Laboratoire d'Immunorhumatologie Moléculaire, Plateforme GENOMAX, INSERM UMR_S 1109, Faculté de Médecine, Fédération Hospitalo-Universitaire OMICARE, ITI TRANSPLANTEX NG, Université de Strasbourg, Strasbourg, France;   [4]Strasbourg Federation of Translational Medicine (FMTS), Strasbourg University, Strasbourg, France;   [5]Institut de Génétique et de Biologie Moléculaire et Cellulaire, INSERM U1258/CNRS UMR7104, Illkirch, France;   [6]Sorbonne Université, INSERM UMR_S 1166 ICAN, Paris, France;   [7]Service d'Immunologie Biologique, Plateau Technique de Biologie, Pôle de Biologie, Nouvel Hôpital Civil, Hôpitaux Universitaires de Strasbourg, Strasbourg, France;   [8]Department of Dermatology, Venereology and Allergology, Medical University of Innsbruck, Innsbruck, Austria;   [9]Institute for Molecular Medicine and Paul Klein Center for Immunotherapy (PKZI), University Medical Center of the Johannes Gutenberg-University Mainz, Mainz, Germany.

*F. Arbogast and R. Sal-Carro contributed equally to this paper;   **F. Gros and V. Flacher contributed equally to this paper.   Correspondence to Frédéric Gros: f.gros@unistra.fr;   Vincent Flacher: v.flacher@ibmc-cnrs.unistra.fr

Q. Frenger's and F. Gros's current affiliation is INSERM UMR_S 1109 Immunorhumatologie Moléculaire, Fédération de Médecine Translationnelle de Strasbourg (FMTS), ITI Transplantex NG, Centre de Recherche en Biomédecine de Strasbourg (CRBS), Faculté des Sciences de la Vie, Université de Strasbourg, Strasbourg, France.

2018; Clarke and Simon, 2018). The core autophagy proteins are encoded by autophagy-related (*Atg*) genes. Autophagy is promoted under energetic stress notably through the inhibition of the PI3K/Akt/mTOR pathway (Galluzzi et al., 2014). Autophagy also contributes to metabolic equilibrium in homeostatic conditions as it is a key process in supporting energy provision. For cells relying on oxidative phosphorylation to generate ATP, autophagy contributes to maintaining a functional mitochondrial pool through the degradation of defective mitochondria and in the mobilization of fatty acids through the degradation of lipid droplets in a process called lipophagy (Zhang et al., 2022a). To support homeostasis, autophagy also acts as a quality-control mechanism during the unfolded protein response (UPR), preventing the accumulation of misfolded protein aggregates and degrading excess or damaged endoplasmic reticulum (ER) (Anding and Baehrecke, 2017). These housekeeping forms of autophagy are particularly important in long-lived and self-renewing cells. In the immune system, B-1 B cells, memory B, and T cells as well as plasma cells rely on autophagy for their maintenance (Arbogast et al., 2018; Arnold et al., 2016; Clarke et al., 2018; Murera et al., 2018; Pengo et al., 2013; Xu et al., 2014). ATG proteins are also involved in several non-autophagic processes such as LC3-associated phagocytosis (LAP). LAP requires Rubicon (*Rubcn*) to form an initiation complex and is involved in microorganism clearance, efferocytosis, and antigen presentation, which are highly relevant for DCs and macrophages (Heckmann and Green, 2019; Münz, 2015). Notably, autophagy impairment in DCs notably leads to defective CD4+ and CD8+ T cell responses (Alissafi et al., 2017; Lee et al., 2010; Mintern et al., 2015; Weindel et al., 2017).

Overall, selective deletion of *Atg* genes in macrophages and DCs has demonstrated that autophagy modulates pathogen resistance, antigen presentation, and proinflammatory signals, i.e., inflammasome activity (Bah and Vergne, 2017; Ghislat and Lawrence, 2018; Valečka et al., 2018; Takahama et al., 2018). Similarly, recent reports support the role of autophagy for LCs in the regulation of inflammatory responses (Müller et al., 2020; Said et al., 2014) and in the immune response against intracellular bacteria (Dang et al., 2019). Moreover, autophagy proteins participate in the intracellular routing of human immunodeficiency virus (HIV) particles toward degradative compartments in human LCs upon Langerin/CD207-mediated uptake (Ribeiro et al., 2016). Interestingly, enhancing autophagy by pharmacological agents limits HIV-1 mucosal infection and replication (Cloherty et al., 2021). Yet, non-autophagic roles of ATG proteins cannot be ruled out to explain these results.

When autophagy defects were assessed in vivo for cDCs and macrophages, there was no report of impaired cell survival (Lee et al., 2010; Mintern et al., 2015; Oh and Lee, 2019; Wu and Lu, 2019), except for a peritoneal macrophage subset (Xia et al., 2020) Although some of the conditional deletion systems used for these investigations may have impacted LCs as well, no information is currently available on the consequences of constitutive autophagy impairment for their maintenance in vivo. Since LCs are self-renewing, long-lived APCs that are exposed to low availability of nutrients, UV irradiation, or stress related to infection, we hypothesized that efficient autophagy might be a

key element supporting their maintenance in the epidermis. To investigate this, we generated *Cd207*-specific deletion of *Atg5* to define primary roles of autophagy and related processes in LC biology.

## Results

### ATG5 is necessary for Langerhans cell network maintenance

Since evidence for autophagosomes in primary LCs has been so far limited (Ribeiro et al., 2016), we first verified whether LCs from digested murine epidermis comprise such compartments. Electron microscopy of LCs, including original images and re-analysis of previously published samples (Schuler and Steinman, 1985), allowed the identification of double-membrane compartments as well as crescent-shape structures reminiscent of incipient phagophores and isolation membranes. The diameter of the autophagosomes was between 400 and 600 nm (Fig. S1). In line with this observation, immunofluorescence revealed LC3-positive compartments within LCs (Fig. 1 A, *Atg5^{WT}*). When LCs were treated with hydroxychloroquine to block the lysosomal degradation of autophagosomes, we observed an accumulation of membrane-associated LC3 by flow cytometry, thereby demonstrating autophagic flux (Fig. 1 B, *Atg5^{WT}*). Of note, quantification of the flux gave similar results in freshly isolated, immature LCs, and in mature LCs migrating out of cultured epidermal sheets (Fig. 1 C, *Atg5^{WT}*). Thus, LCs of wild-type mice display constitutive autophagic activity, regardless of their maturation state.

To determine the function of autophagy in LCs in vivo, we generated *Cd207-cre x Atg5^{flox/−}* (*Atg5^{ΔCd207}*) mice, in which the essential autophagy gene *Atg5* is deleted by CRE-mediated recombination in cells expressing CD207 (Fig. S2 A). The efficiency of the deletion was verified by RT-qPCR of LCs sorted from the mouse epidermis (Fig. S2 B) and from skin-draining LNs of 4-wk-old mice (Fig. S2, C and D). This confirmed that the breeding strategy resulted in an optimal deletion efficiency, as *Atg5* mRNA was undetectable in LCs from *Atg5^{ΔCd207}* mice, as compared with LCs from *Atg5^{flox/+}* and *Cd207-cre x Atg5^{flox/+}* control animals (respectively referred to as *Atg5^{WT}* and *Atg5^{WT/Δ}* below). With respect to migratory dermal DCs isolated from LNs of *Atg5^{ΔCd207}* mice, *Atg5* mRNA was absent from CD103+ dermal cDC1, which also express CD207 (Henri et al., 2010), but still present in CD207− MHCII^{high} dermal DCs (Fig. S2 D).

To address whether the *Atg5* deletion leads to autophagy impairment in LCs, the formation of autophagic compartments was assessed by LC3 immunostaining. LC3+ punctate staining in the cytoplasm of *Atg5^{WT}* LCs was clearly visible, whereas LC3 staining was diffuse in LCs from *Atg5^{ΔCd207}* mice (Fig. 1 A; and Videos 1 and 2). This reflects the expected consequences of ATG5 deficiency, i.e., the absence of LC3 conjugation with phosphatidylethanolamine (LC3-II) and lack of integration into autophagic compartments. Finally, we observed that autophagic flux was abolished in both immature and mature LCs of *Atg5^{ΔCd207}* mice (Fig. 1 C). This shows that ATG5 deletion leads to autophagy impairment in LCs.

To monitor the possible involvement of autophagy in the homeostatic maintenance of LCs under steady state conditions,

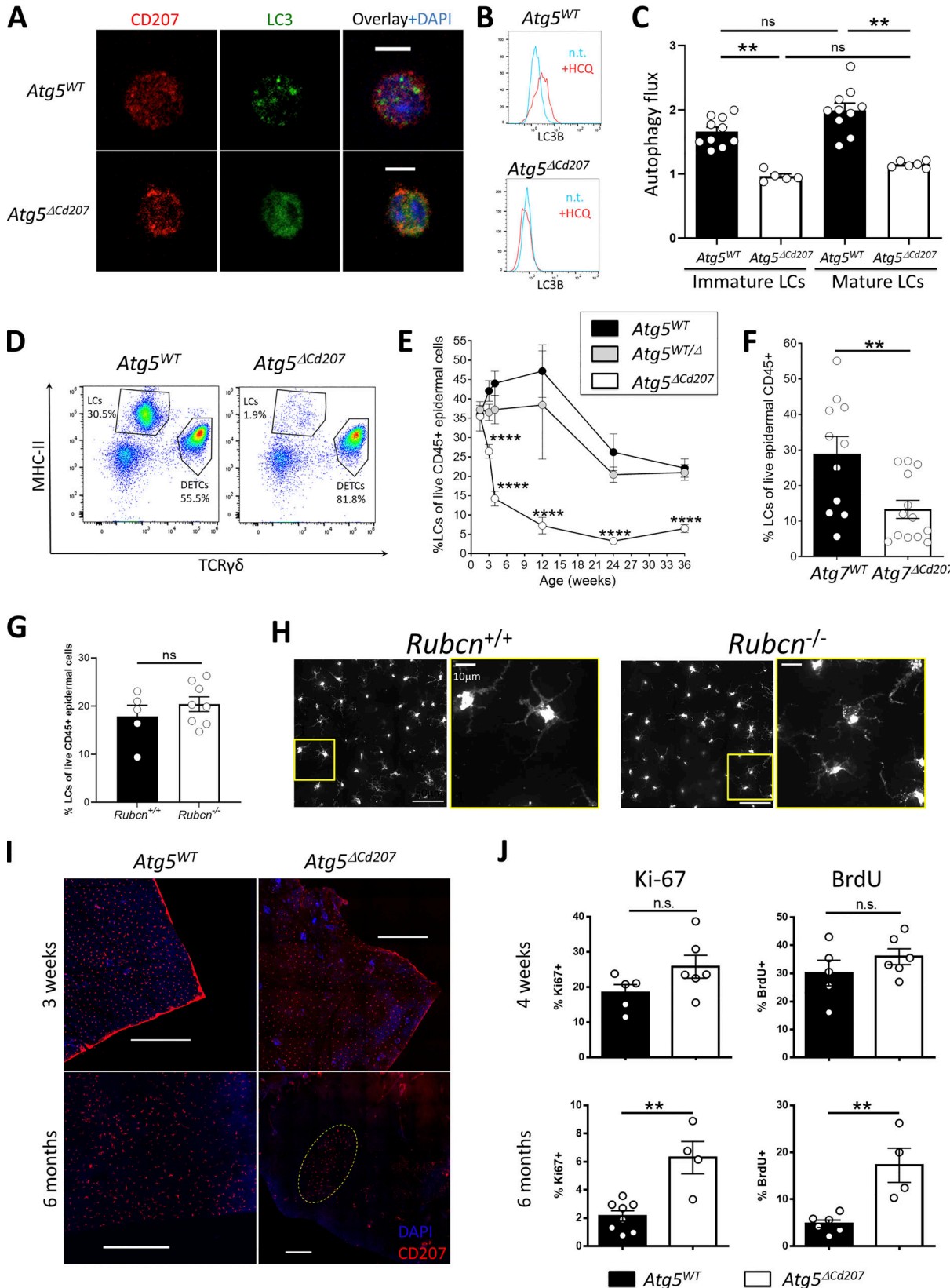

Figure 1. **ATG5 deficiency in Langerhans cells disrupts autophagosomes and depletes their epidermal network. (A)** Representative immunofluorescent staining of MAP1LC3B (LC3) on LCs obtained by in vitro emigration from epidermal sheets of $Atg5^{WT}$ (Video 1) and $Atg5^{\Delta Cd207}$ (Video 2) mice. LC3: green; CD207: red; DAPI: blue. Scale bars: 10 μm. **(B and C)** Representative histogram plot of LC3B staining and (C) quantification of autophagy flux in immature or mature LCs of 3-wk-old $Atg5^{WT}$ and $Atg5^{\Delta Cd207}$ mice, treated or not with chloroquine. Autophagy flux was calculated as a ratio between MFI for LC3B in treated and untreated cells. **(D)** Representative dot plots for the identification of MHCII+ TCRγδ− Langerhans cells (LCs; all CD207+) and MHCII− TCRγδ+ dendritic epidermal

T Cells (DETCs) among CD45+ cells in freshly digested back skin epidermal suspension of 6-mo-old $Atg5^{WT}$ and $Atg5^{\Delta Cd207}$ mice. **(E)** Comparison over time of the percentage of LCs among live CD45+ epidermal cells for control ($Atg5^{WT}$ and $Atg5^{WT/\Delta}$) and $Atg5^{\Delta Cd207}$ mice. **(F)** Percentage of LCs among live CD45+ cells in freshly digested back skin epidermis of 10–20 wk-old $Atg7^{WT}$ and $Atg7^{\Delta Cd207}$ mice. **(G)** Percentage of CD45+ MHCII+ CD207+ LCs among live CD45+ cells obtained from fresh epidermal cell suspensions of 2-mo-old $Rubicn^{-/-}$ and $Rubicn^{+/+}$ mice. **(H)** Immunofluorescence staining of CD207+ LCs in ear epidermal sheets of 2-mo-old $Rubicn^{-/-}$ and $Rubicn^{+/+}$ mice. Images are representative of two independent experiments. Insets 1 and 2: close-up views. **(I)** Representative immunofluorescent staining of CD207 on epidermal sheets of ear skin from 3 wk (upper panels) and 6-mo-old (lower panels) $Atg5^{WT}$ and $Atg5^{\Delta Cd207}$ mice. Scale bars: 100 μm. **(J)** Percentages of epidermal LCs stained for BrdU incorporation and Ki67 expression for 4-wk (top) and 6-mo-old (bottom) $Atg5^{WT}$ and $Atg5^{\Delta Cd207}$ mice. All data are pooled from at least three independent experiments, with each point representing one individual mouse (except [E]: $n \geq 4$ mice per time-point). Statistical analysis: Mann–Whitney U test (except [C]: Kruskal–Wallis one-way ANOVA followed by Dunn's multiple comparison test; and [E]: two-way ANOVA followed by Tukey's multiple comparison test). **, $P < 0.01$; ***, $P < 0.001$; ****, $P < 0.0001$; ns, $P > 0.05$.

we evaluated their epidermal network at different ages. Since CD207 expression in MHCII+ epidermal LC precursors is completed around 7–10 days after birth (Tripp et al., 2004), we assessed the proportion of MHCII+ TCRγδ− CD207+ LCs among CD45+ epidermal cells by flow cytometry from 10 days until 9 mo of age (Fig. 1 D). The basal proportion of LCs at 10 days was comparable for mice of all genotypes, suggesting that no major defect occurs in the seeding of MHCII+ CD207− embryonic LC precursors in the epidermis, which also corresponds to the expected kinetics of $Cd207$ promoter activity and CRE expression (Fig. 1 E). In $Atg5^{WT}$ and $Atg5^{WT/\Delta}$ mice, we observed a moderate increase of LCs until 6–12 wk, followed by a decrease in aging mice. Nevertheless, the proportion of LCs diminished sharply around 2–4 wk of age in the epidermis of $Atg5^{\Delta Cd207}$, eventually stabilizing at around 5% of epidermal CD45+ leukocytes at 9 mo.

To reinforce the conclusion that the loss of LCs is due to impaired autophagy and not other ATG5-related cellular homeostatic dysfunctions, we generated $Cd207$-cre x $Atg7^{flox/flox}$ ($Atg7^{\Delta Cd207}$) mice and compared their epidermal cell suspensions with that of $Atg7^{flox/flox}$ ($Atg7^{WT}$) mice. Similar to our findings with $Atg5^{\Delta Cd207}$ mice, ATG7 deficiency resulted in a significant depletion of LCs from the epidermis of mice older than 10 wk (Fig. 1 F). Thus, we can exclude effects only linked to ATG5, such as direct regulation of apoptosis independently of the autophagy machinery (Galluzzi et al., 2014). Additionally, we could exclude a role for LAP or other endocytic processes requiring ATG5 as the density of the epidermal network of LCs appeared unaffected in Rubicon-deficient mice (Fig. 1, G and H).

We next performed immunofluorescent staining of the LC network in epidermal sheets prepared from the ear skin of $Atg5^{\Delta Cd207}$ and control mice (Fig. 1 I). We did not observe any obvious difference in the LC network in mice, regardless of their genotype. However, and in accordance with flow cytometry results, very few LCs were visible in 6-mo-old $Atg5^{\Delta Cd207}$ mice. LCs remaining in older mice were often assembled in disseminated patches. This pattern is reminiscent of the network reconstitution that occurs through slow in situ LC proliferation following induced depletion in Langerin-DTR mice (Bennett et al., 2005). Indeed, LCs ensure the integrity of their epidermal network by self-renewal (Chorro et al., 2009; Ghigo et al., 2013). Consequently, we assessed the proliferative capacity of ATG5-deficient LCs by 5-bromo-2′-deoxyuridine (BrdU) incorporation and Ki-67 staining by flow cytometry (Fig. 1 J). We observed proliferation rates consistent with our previous observations in 4-wk-old mice (Voisin et al., 2019, Preprint), with comparable percentages of BrdU+ and Ki-67+ LCs in $Atg5^{\Delta Cd207}$ and control $Atg5^{WT}$ mice, thereby concluding that

autophagy deficiency does not prevent maintenance of the LC network by a major proliferative impairment. On the other hand, LCs of 6-mo-old $Atg5^{\Delta Cd207}$ mice displayed higher proliferation rates, presumably because of homeostatic compensation for the depletion of their epidermal niche.

Finally, since dermal cDC1 also expressed CD207 and showed a decrease of $Atg5$ transcripts in $Atg5^{\Delta Cd207}$ mice (Fig. S2 D), we quantified LC and cDC1 populations among total MHCII+ CD11c+ skin DCs (Fig. S2 E). We confirmed the significant decrease of LCs present in whole skin cell suspensions, while the proportion of cDC1 among skin DCs was rather slightly increased (Fig. S2 F). Thus, the core autophagic machinery is dispensable for the maintenance of dermal cDC1, yet appears essential to LCs.

**ATG5-deficient Langerhans cells undergo limited apoptosis**
Since their self-renewal was not affected, the loss of ATG5-deficient LCs might be explained by an enhanced migration into lymph nodes or by increased cell death. LCs, similar to cDCs of peripheral tissues, undergo maturation and migrate to skin-draining LNs following inflammatory signals (Doebel et al., 2017; Kaplan, 2017). Alternatively, spontaneous maturation of LCs may result from disrupted TGF-β signaling, which, under physiological conditions, is required to maintain an immature state (Bobr et al., 2012; Kel et al., 2010). In both cases, an increased expression of maturation markers MHCII and CD86 can be observed prior to the departure of LCs from the epidermis. We thus verified whether autophagy impairment might prompt spontaneous LC maturation. MHCII and CD86 expression by LCs in the epidermis did not show any variation in $Atg5^{\Delta Cd207}$ compared with control mice (Fig. 2, A and B). Moreover, because an overt LC emigration from the epidermis would lead to a noticeable accumulation in LNs, we determined LC numbers in inguinal and brachial LNs of 6-wk-old mice. We observed instead a trend toward a decrease, only significant in percentage for LCs from $Atg5^{\Delta Cd207}$ mice (Fig. 2, C and D). A similar pattern was observed for dermal cDC1 that also expresses CD207. In contrast, no differences were detected for CD207− dermal DC subsets that lack $Atg5$ deletion. Taken together, these results exclude that impaired autophagy leads to a massive spontaneous maturation and migration of LCs.

Finally, using flow cytometry, we addressed whether the absence of $Atg5$ might lead to increased apoptosis by measuring the proportion of LCs with active caspase-3. This major effector of apoptosis was detected in a small, yet significant proportion of ATG5-deficient LCs, both in the epidermis and LNs (Fig. 2 E). Altogether, these results demonstrate that reduced cell division or increased maturation and migration cannot account for LC

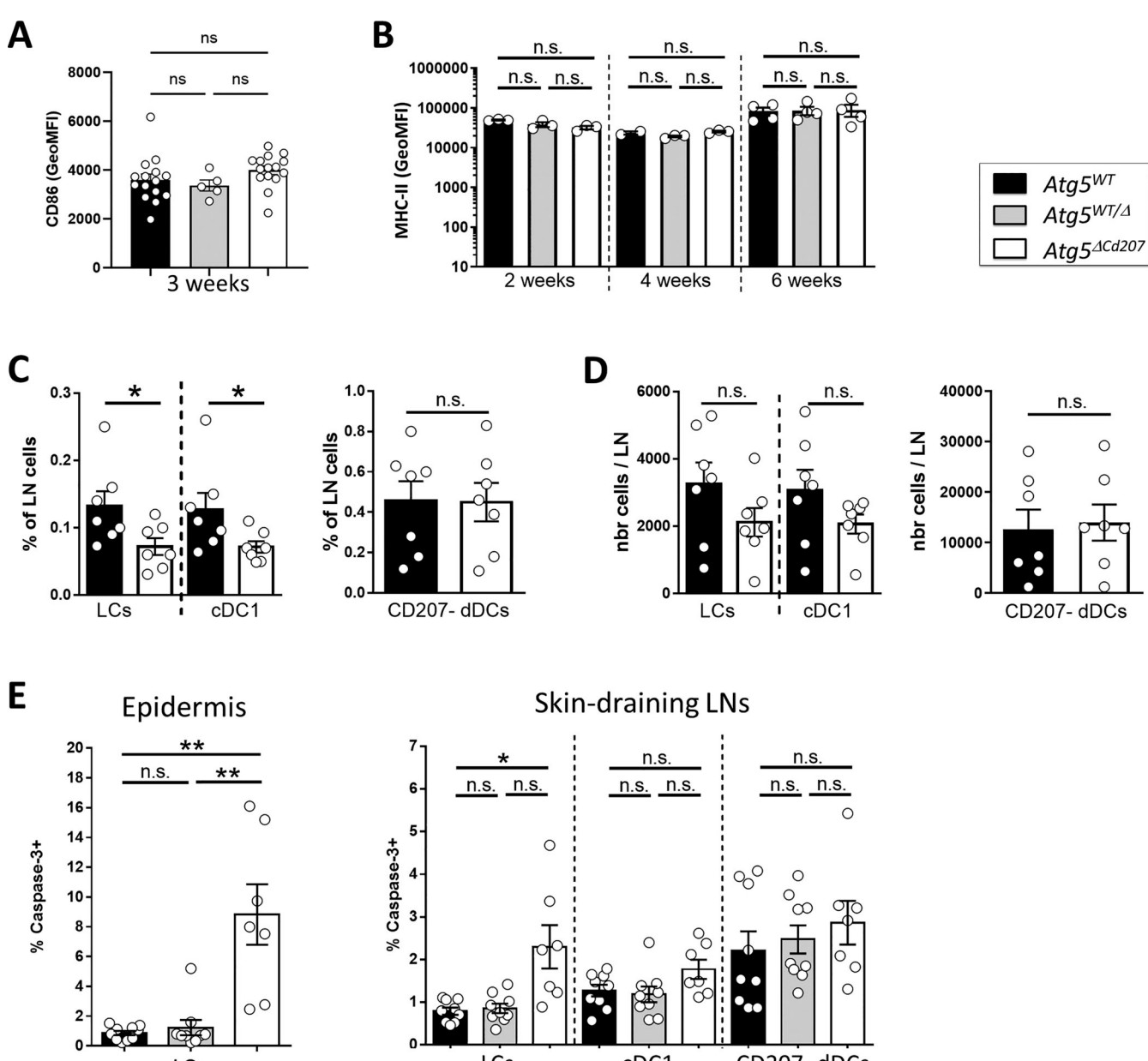

Figure 2. **ATG5-deficient Langerhans cells remain immature and undergo apoptosis. (A and B)** CD86 (A) and MHCII (B) MFI for epidermal LCs of 3-wk-old control (*Atg5^WT* and *Atg5^WT/Δ*) and *Atg5^ΔCd207* mice. Data are pooled from five independent experiments, with each point representing one individual mouse. Statistical analysis: Kruskal–Wallis one-way ANOVA followed by Dunn's multiple comparison test (ns, P > 0.05). **(C and D)** Percentages (C) and absolute numbers (D) of LCs, cDC1, and CD207⁻ DCs in freshly digested skin draining lymph nodes of 3-wk-old control (*Atg5^WT* and *Atg5^WT/Δ*) and *Atg5^ΔCd207* mice. Data are pooled from at least three independent experiments, with each point representing one individual mouse. Statistical analysis: Mann–Whitney U test (*, P < 0.05; ns, P > 0.05). **(E)** Percentage of cells with activated caspase-3 in LCs of freshly digested back skin epidermis (left panel) and LCs, cDC1, and CD207- dermal DCs of skin-draining lymph nodes (right panel) for 3-wk-old control (*Atg5^WT* and *Atg5^WT/Δ*) and *Atg5^ΔCd207* mice. Data are pooled from at least three independent experiments, with each point representing one individual mouse. Statistical analysis: Kruskal–Wallis one-way ANOVA followed by Dunn's multiple comparison test (*, P < 0.05; **, P < 0.01; ns, P > 0.05).

network disintegration, whereas apoptosis of ATG5-deficient LCs, albeit limited, suggests autophagy as crucial, even in the steady state, for LC survival.

**ATG5-deficient Langerhans cells show endoplasmic reticulum swelling but no unfolded protein response**
Functional autophagy is required for the maintenance of the endoplasmic reticulum (ER). ER damage triggers the inositol-

requiring enzyme 1 (IRE1)/X-box binding protein 1 (Xbp1) axis of the UPR, which is a master regulator of DC survival and maturation (Cubillos-Ruiz et al., 2015; Grootjans et al., 2016; Tavernier et al., 2017) but has not been investigated in LCs so far. Autophagy regulates ER swelling, protein aggregation, and thereby limits the extent of the UPR (Song et al., 2018). In line with this, exposure of wild-type LCs to the phosphatidylinositol-3-kinase inhibitor, wortmannin, which inhibits the initiation of

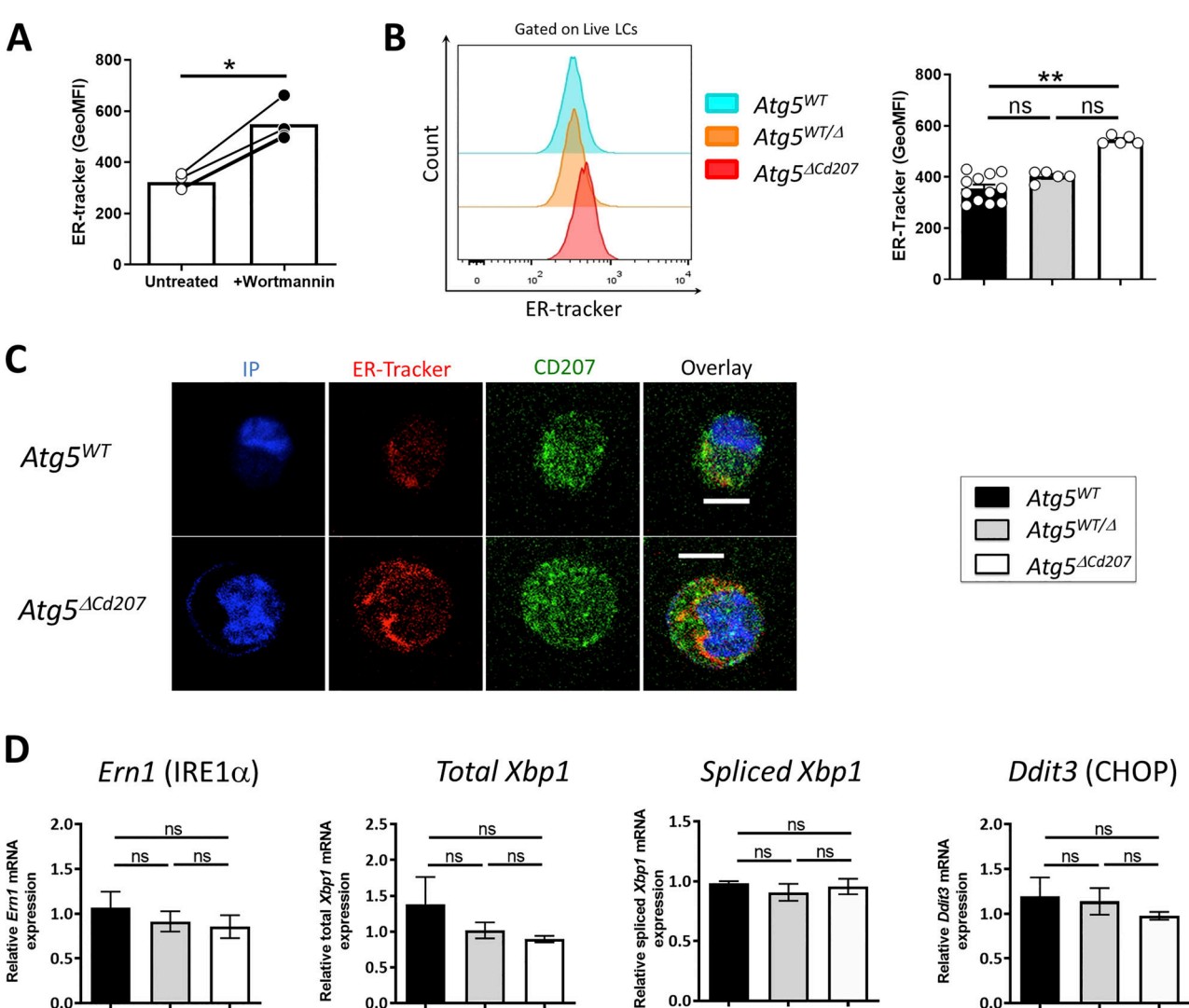

Figure 3. **ATG5-deficient Langerhans cells present endoplasmic reticulum swelling but no unfolded protein response. (A)** MFI for ER-tracker on epidermal LCs treated or not with Wortmannin. Data are pooled from at least three independent experiments, with each point representing one individual 3-wk-old mouse. Statistical analysis: Mann–Whitney U test (*, P < 0.05). **(B)** Representative half-set overlay (left panel) and MFI (right panel) of ER-tracker in epidermal LCs of control (*Atg5^WT* and *Atg5^WT/Δ*) and *Atg5^ΔCd207* mice. Data are pooled from at least three independent experiments, with each point representing one individual 3-wk-old mouse. Statistical analysis: Kruskal–Wallis one-way ANOVA followed by Dunn's multiple comparison test (**, P < 0.01). **(C)** Representative immunofluorescent staining of the endoplasmic reticulum using the ER-tracker dye on epidermal LCs of 3-wk-old *Atg5^WT* (Video 3) and *Atg5^ΔCd207* (Video 4) mice. CD207: green; ER-tracker: red; DAPI: blue. Scale bar: 10 μm. **(D)** Expression of *Ern1*, total and spliced *Xbp1*, and *Ddit3* mRNAs, in epidermal LCs of 3-wk-old control (*Atg5^WT* and *Atg5^WT/Δ*) and *Atg5^ΔCd207* mice. Fold changes were calculated relative to mRNA expression in cells of *Atg5^WT* control mice. Data are pooled from at least three independent experiments, with each point representing one individual mouse. Statistical analysis: Kruskal–Wallis one-way ANOVA followed by Dunn's multiple comparison test (ns, P > 0.05).

the autophagosome formation, resulted in an increased labelling by ER-tracker, a fluorescent dye specific for ER membranes (Fig. 3 A). Then, we stained *Atg5*-deficient LCs from 3-wk-old mice with ER-tracker. Flow cytometry analysis showed a significantly increased ER-tracker staining in LCs of *Atg5^ΔCd207* compared to wild-type mice (Fig. 3 B). In line with this, confocal microscopy revealed an expanded ER compartment in these cells (Fig. 3 C; and Videos 3 and 4). These signs of ER expansion prompted us to study whether the expression of key intermediates of the UPR pathway might be elevated. Quantitative PCR was performed for *Ern1*, total *Xbp1*, spliced *Xbp1*, and *Ddit3* mRNA. However, none of these genes showed increased

expression, demonstrating that the UPR pathway was not constitutively engaged (Fig. 3 D). Therefore, we conclude that ATG5-deficient LCs can cope with the observed ER swelling, which does not trigger a massive UPR that could lead to cell death.

**Autophagy-deficient Langerhans cells accumulate intracellular lipid storage**

To identify dysregulated gene expression patterns that could be linked with impaired autophagy, RNA sequencing was performed on epidermal LCs sorted from 3-wk-old *Atg5^ΔCd207* or *Atg5^WT* control mice. Analysis of *Atg5* mRNA sequencing reads

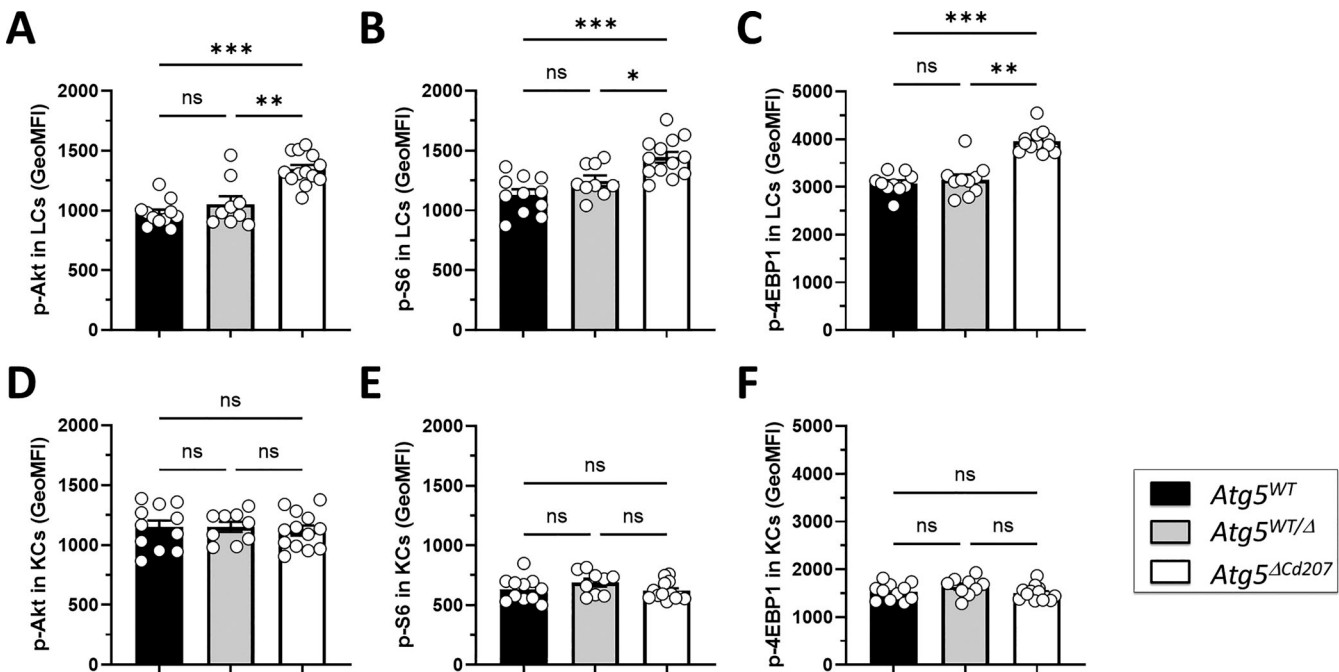

Figure 4. **The PI3K/Akt pathway is promoted in ATG5-deficient Langerhans cells. (A–F)** MFI of (A and D) phosphorylated Akt, (B and E) phosphorylated S6, and (C and F) phosphorylated 4E-BP1 in epidermal mature LCs (A–C) or KCs (D–F) obtained from control ($Atg5^{WT}$ and $Atg5^{WT/\Delta}$) and $Atg5^{\Delta Cd207}$ mice, quantified by flow cytometry. Data are pooled from three independent experiments, with each point representing one individual 3-wk-old mouse. Statistical analysis: Kruskal–Wallis one-way ANOVA followed by Dunn's multiple comparison test (*, $P < 0.05$; **, $P < 0.01$; ***, $P < 0.001$; ns, $P > 0.05$).

confirmed the deletion of exon 3 in LCs upon CRE-mediated recombination (Fig. S3 A). Principal component analysis revealed clear differences in transcriptomic profiles between $Atg5^{\Delta Cd207}$ and $Atg5^{WT}$ mice (Fig. S3 B). Differentially expressed genes in $Atg5^{\Delta Cd207}$ LCs included 673 upregulated and 629 downregulated genes (Table S1; and Fig. S3, C and D). Gene ontology pathway enrichment analysis suggested in particular a dysregulation of cellular metabolism (GO:0046942, GO:0043269, GO:0017144, GO:0044272, GO:0051186, GO:0110096, GO:0046085, GO:1901615, GO:0015711, GO:0009166, and GO:0007584; Fig. S3 E).

The PI3K/Akt pathway plays a crucial role in the regulation of autophagy. We identified that ATG5-deficient LCs decreased their expression of Inositol-3-phosphate synthase 1 (*Isyna1*) and of TNF-α-induced protein 8-like protein 3 (*Tnfaip8l3*), which shuttles PIP2 and PIP3 across the plasma membrane (Fayngerts et al., 2014), and upregulated transcription of Protein Kinase C beta (*Prkcb*) (Table S1). Altogether, this may be interpreted as compensation for PI3K/Akt hyperactivation. Indeed, autophagy-deficient LCs, but not keratinocytes from the same mice, exhibited a constitutive activation of the PI3K pathway, as demonstrated by the increased phosphorylation of Akt, S6, and 4E-BP1 (Fig. 4).

Autophagy regulates cellular lipid metabolism by lipophagy, which has a crucial role in balancing energy supply in both steady state and under metabolic stress. Lipophagy mediates lysosomal degradation of proteins that coat cytoplasmic lipid droplets, and lipolysis of triglycerides, thus liberating free fatty acids to be consumed by beta-oxidation in mitochondria (Kounakis et al., 2019). Accordingly, LCs of $Atg5^{\Delta Cd207}$ mice

modulated the expression of several genes encoding actors of lipidic metabolism pathways (Fig. 5 A). We noticed upregulation of mRNA of the solute carrier (SLC) family transporters MCT-4/SLC16A3 (lactate), SLC7A11 (cysteine, glutamate), and SLC7A2 (lysine, arginine), which import molecules that directly or indirectly provide substrates to the tricarboxylic acid cycle. Upregulated expression of Acyl-CoA Synthetase Short Chain Family Member 1 and 2 (*Acss1* and *Acss2*) is expected to favor the synthesis of Acetyl-CoA, which can either be converted into lipids or fuel mitochondrial beta-oxidation. Fatty acid synthesis and energy storage in the form of lipid droplets appear to be favored in ATG5-deficient LCs, as hinted by the upregulation of *Gyk*, encoding the glycerol kinase which catalyzes triglyceride synthesis, and *Acsl3*, encoding the Acyl-CoA Synthetase Long-Chain Family Member 3, a key enzyme for neutral lipid generation (Gao et al., 2019).

Using flow cytometry, we noticed an increased side scatter of $Atg5^{\Delta Cd207}$ LCs (Fig. 5 B). Considering that lipid metabolism appeared deregulated in autophagy-deficient LCs, we hypothesized that this increased granularity may be due to a larger number of intracellular lipid-rich vesicles. Thus, epidermal cell suspensions were exposed to Bodipy, a lipid staining dye that targets neutral lipid-rich compartments, which we first quantified by flow cytometry. We first validated this experimental approach by treating wild-type LCs with etomoxir, a carnitine palmitoyltransferase I inhibitor that blocks the import of activated free fatty acids (acyl-CoA) by mitochondria. Etomoxir treatment resulted in a stronger intensity of Bodipy staining, reflecting higher neutral lipid storage by LCs (Fig. 5 C). Treating LCs with the autophagy inhibitor wortmannin also resulted in an

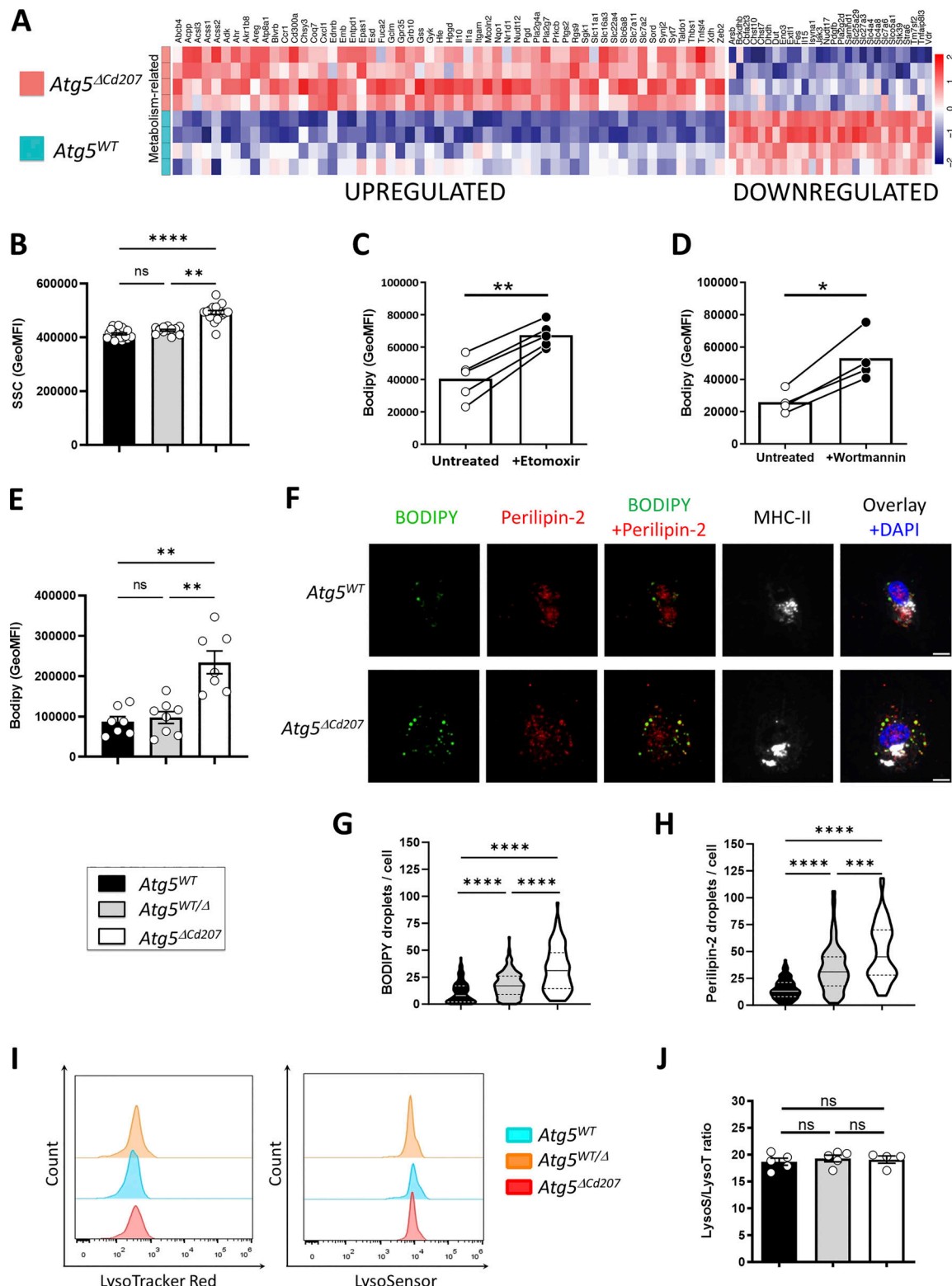

Figure 5. **Impaired autophagy increases the lipid storage compartments of Langerhans cells. (A)** Differentially expressed transcripts related to lipid metabolism pathways in *Atg5^WT* versus *Atg5^ΔCd207* LCs (GO:0046942, GO:0043269, GO:0017144, GO:0044272, GO:0051186, GO:0110096, GO:0046085, GO: 1901615, GO:0015711, GO:0009166, and GO:0007584). **(B)** Flow cytometry quantification of the side scatter (SSC) MFI in epidermal LCs obtained from control (*Atg5^WT* and *Atg5^WT/Δ*) and *Atg5^ΔCd207* mice. Data are pooled from 10 independent experiments, with each point representing one individual 3-wk-old mouse. Statistical analysis: Kruskal–Wallis one-way ANOVA followed by Dunn's multiple comparison test (**, P < 0.01; ****, P < 0.0001; ns, P > 0.05). **(C and D)** Flow cytometry quantification of the Bodipy MFI in epidermal LCs obtained from C57BL/6 mice then treated with (C) etomoxir or (D) wortmannin. Data are pooled from at least three independent experiments, with each point corresponding to one individual 3-wk-old mouse. Statistical analysis: Mann–Whitney U test (*, P < 0.05). (*, P < 0.05; **, P < 0.01). **(E)** Flow cytometry quantification of Bodipy MFI in epidermal LCs obtained from control (*Atg5^WT* and *Atg5^WT/Δ*) and *Atg5^ΔCd207* mice and

*Atg5^{ΔCd207}* mice. Data are pooled from four independent experiments, with each point representing one individual 3-wk-old mouse. Statistical analysis: Kruskal–Wallis one-way ANOVA followed by Dunn's multiple comparison test (**, P < 0.01; ns, P > 0.05). **(F)** Immunofluorescent staining of MHCII+ epidermal LCs obtained from *Atg5^{WT}* (upper panels) and *Atg5^{ΔCd207}* (lower panels) mice and stained with Bodipy or anti-Perilipin-2 antibody. Scale bars: 10 µm. **(G and H)**. Quantification of (G) Bodipy+ and (H) Perilipin-2+ vesicles in LCs obtained from control (*Atg5^{WT}* and *Atg5^{WT/Δ}*) and *Atg5^{ΔCd207}* mice. Data are representative of three independent experiments and presented as violin plots (solid line, median; dashed lines, first and third quartiles; n > 100 cells from a total of three 3-wk-old mice). Statistical analysis: Kruskal–Wallis one-way ANOVA followed by Dunn's multiple comparison test ***, P < 0.001; ****, P < 0.0001). **(I)** Representative half-set overlay of LysoTracker-Red (left panel), and LysoSensor (right panel) staining. **(J)** Comparison of the ratio of MFI of each marker for epidermal LCs of control (*Atg5^{WT}* and *Atg5^{WT/Δ}*) and *Atg5^{ΔCd207}* mice. Data are pooled from at least three independent experiments, with each point corresponding to one 3-wk-old individual mouse. Statistical analysis: Kruskal–Wallis one-way ANOVA followed by Dunn's multiple comparison test (ns, P > 0.05).

increased Bodipy staining, suggesting constitutive lipophagy in LCs (Fig. 5 D). We then found that *Atg5^{ΔCd207}* LCs retained more Bodipy as compared with the LCs of control mice (Fig. 5 E). We consistently observed by confocal microscopy that LCs of *Atg5^{ΔCd207}* mice contained more Bodipy-positive vesicular structures that could correspond to lipid droplets (Fig. 5, F and G; and Videos 5 and 6). To unequivocally identify this lipid storage compartment, we were interested in perilipins, a family of lipid droplet–specific proteins. According to Immgen gene expression datasets (Miller et al., 2012), *Plin2*, encoding Perilipin-2, is the most strongly expressed in LCs among this family. Staining for Perilipin-2 lipid droplets confirmed their accumulation in autophagy-deficient LCs (Fig. 5, F and H). Both Bodipy+ and Perilipin-2+ vesicles were already increased in LCs of *Atg5^{WT/Δ}* mice as compared with *Atg5^{WT}* mice (Fig. 5, G and H), suggesting that even a moderate impairment of autophagy results in an abnormal increase of neutral lipid storage compartments. Of note, the enlargement of ER (Fig. 3) may be linked to the increased production of lipid droplets, which originate from this compartment (Zadoorian et al., 2023). Finally, since this accumulation of lipid droplets may be the consequence of impaired degradation of autophagosomes, the acidification and lysosomal load were quantified by Lysosensor and Lysotracker probes, respectively (Fig. 5 I). We could thus verify that lysosomes were unperturbed in the absence of ATG5 (Fig. 5 J).

## Disrupted lipid metabolism in autophagy-deficient Langerhans cells

ATG5 deficiency might lead to an accumulation of intracellular lipids if energy production in LCs strongly relies on lipophagy to mobilize these storage units and produce energy by the beta-oxidation pathway (Kounakis et al., 2019). To assess the energy production in LCs, we focused on AMP-activated protein kinase (AMPK), a master regulator of energetic metabolism, which is phosphorylated on residues T183/T172 when ATP/AMP ratios decline (Herzig and Shaw, 2018). As measured by flow cytometry, AMPK phosphorylation was indeed increased in LCs from *Atg5^{ΔCd207}* mice (Fig. 6 A).

AMPK phosphorylation is expected to promote different pathways that help restore optimal ATP production, including the import of glucose or fatty acid uptake and synthesis. To quantify the glucose uptake intensity, we first monitored the expression of the glucose transporter GLUT-1 by LCs. However, no significant difference could be observed between LCs of *Atg5^{ΔCd207}* mice and control mice (Fig. 6 B). Next, we quantified the glucose uptake of these cells using 2-(N-(7-Nitrobenz-2-oxa-1,3-diazol-4-yl) Amino)-2-Deoxyglucose (2-NBDG), a fluorescent

glucose analog which can be tracked by flow cytometry. In line with the unmodified GLUT-1 expression, autophagy-deficient LCs were not more efficient at capturing glucose than LCs of wild-type mice (Fig. 6 C). On the other hand, LCs from *Atg5^{ΔCd207}* mice exhibited a stronger expression of CD36 (Fig. 6 D). This scavenger receptor has a key role in the capture of free fatty acids and lipids, which was indeed increased when LCs incubated with Bodipy-labeled C16 fatty acid (Fig. 6 E). Altogether, these assays demonstrate that autophagy-deficient LCs show a deficit in energy production, despite an increased ability to capture and store lipids.

Recent insights into different tissue macrophages showed that LCs heavily rely on mitochondrial oxidative phosphorylation (Wculek et al., 2023). To evaluate whether a lack of autophagy affects this pathway, we quantified the metabolic flux of LCs exposed to a series of inhibitors of mitochondrial respiratory complexes (Fig. 6 F). We included bone marrow-derived DCs (BMDCs) because their metabolism has been scrutinized in many previous investigations and relies mostly on beta-oxidation (Pearce and Everts, 2015). The decrease of oxygen consumption rates for wild-type and ATG5-deficient LCs upon oligomycin treatment was similar to that of BMDCs, indicating a similar basal production of ATP (Fig. 6 G: ATP). Wild-type LCs and BMDCs also displayed a comparable profile after exposure to FCCP, which unleashes the maximal respiratory capacity of a cell (Fig. 6 G: Max). On the other hand, LCs of *Atg5^{ΔCd207}* mice reacted to FCCP by a strikingly prominent peak of their oxygen consumption. This implies that, in the absence of autophagy, the potential of LCs to mobilize oxidative phosphorylation, also called spare respiratory capacity, had massively increased (Fig. 6 G: SRC). Despite this, ATG5-deficient LCs appeared unable to use this capacity to promote ATP production (Fig. 6 G: ATP). The respiratory capacity of a cell relies on mitochondria, and transcriptome analysis demonstrated that mitochondria-related genes were differentially regulated upon loss of autophagy (Fig. 6 H). Thus, we performed double staining with mitotracker (MT) Green and Deep Red on LCs extracted from 3-wk-old mice. While MT Deep Red is sensitive to mitochondrial membrane potential, MT Green stains mitochondrial membranes independently of the membrane potential, thus allowing normalization of the membrane potential to the mitochondrial load. An increased mitochondrial mass was detected (Fig. 6 I), in line with the increased capacity of energy production that we measured by the mitochondrial stress assay. ATG5-deficient LCs did not display decreased membrane potential (Fig. 6 J), suggesting that mitochondrial function was preserved. Mitophagy is a key process to eliminate defective mitochondria, in

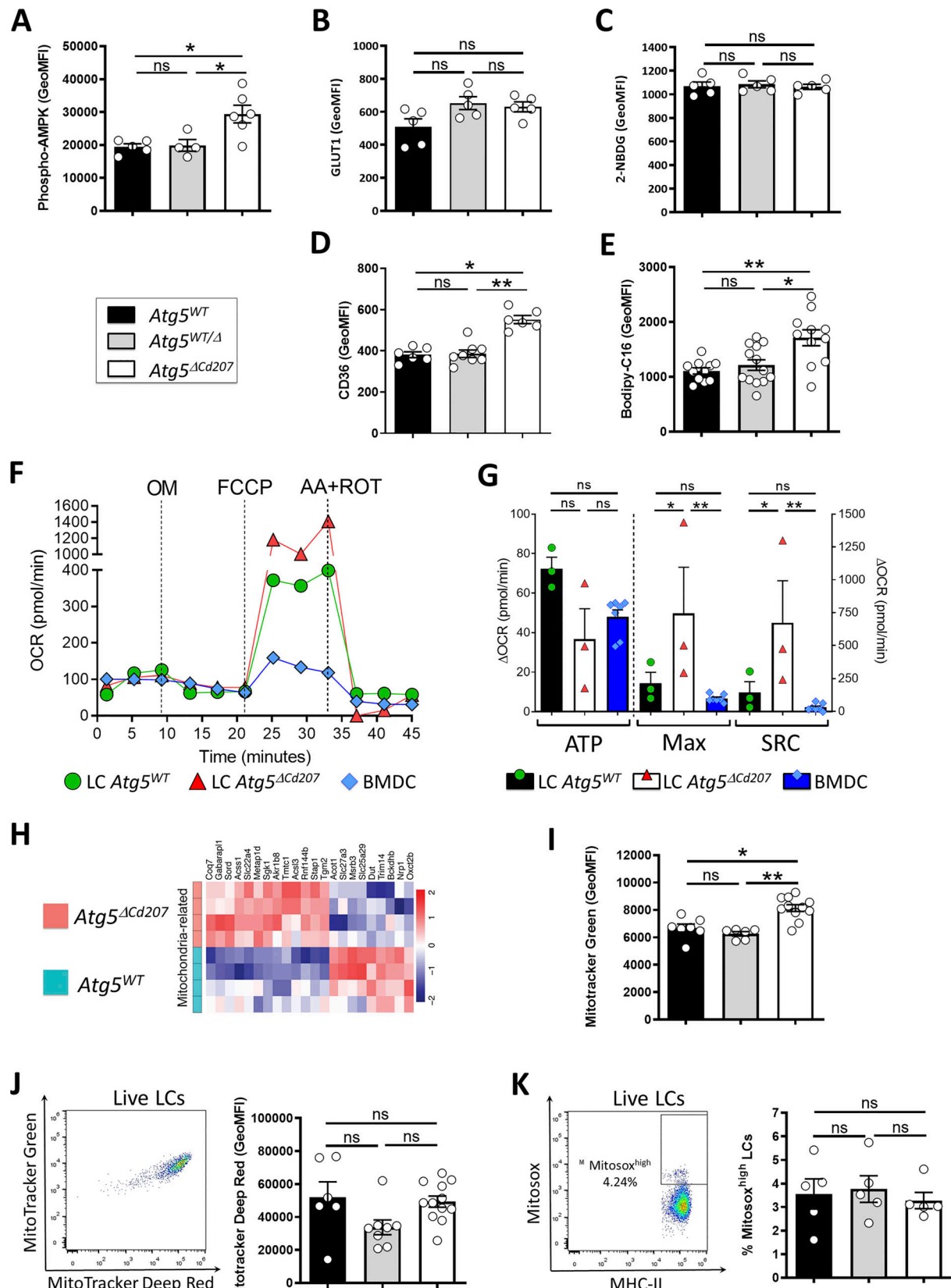

Figure 6. **Impaired lipid metabolism in ATG5-deficient Langerhans cells. (A–E)** Flow cytometry quantification of mean intensity of fluorescence for (A) Phosphorylated AMPK, (B) GLUT1, (C) 2-NDBG uptake, (D) CD36, and (E) Bodipy C16 uptake in epidermal LCs obtained from 3-wk-old control (*Atg5^WT* and *Atg5^WT/Δ*) and *Atg5^ΔCd207* mice. **(F)** Epidermal LCs sorted from *Atg5^WT* or *Atg5^ΔCd207* mice and BMDCs derived from C57BL/6 mice were sequentially exposed to Oligomycin (OM), Carbonyl cyanide 4-(trifluoromethoxy)phenylhydrazone (FCCP), rotenone (ROT) and antimycin A (AA), and oxygen consumption rates (OCR)

were measured by a Seahorse XF96 analyzer throughout the experiment. Data are from one representative experiment out of three. **(G)** ATP production (OCR$_{baseline}$–OCR$_{OM}$), maximum respiration (Max; OCR$_{FCCP}$–OCR$_{AA+ROT}$), and spare respiratory capacity (SRC; OCR$_{FCCP}$–OCR$_{baseline}$) were calculated from the OCR curves. **(H)** Differentially expressed transcripts related to mitochondria (GO: 0005739) in Atg5$^{WT}$ versus Atg5$^{ΔCd207}$ LCs. **(I)** Mitochondrial load for epidermal LCs of Atg5$^{WT}$, Atg5$^{WT/Δ}$, and Atg5$^{ΔCd207}$ mice, as measured by MFI of Mitotracker Green staining. **(J)** Representative dot plot of Mitotracker Green and Deep-Red staining and comparison of Mitotracker Deep-Red MFI of epidermal LCs obtained from control (Atg5$^{WT}$ and Atg5$^{WT/Δ}$) and Atg5$^{ΔCd207}$ mice. **(K)** Representative dot plot of Mitosox Red staining and comparison of Mitosox$^{high}$ percentage of epidermal LCs obtained from control (Atg5$^{WT}$ and Atg5$^{WT/Δ}$) and Atg5$^{ΔCd207}$ mice. All data are pooled from at least three independent experiments, with each point representing one individual 3-wk-old mouse. Statistical analysis: Kruskal–Wallis one-way ANOVA followed by Dunn's multiple comparison test (except g: two-way ANOVA followed by Šídák's multiple comparisons test). (*, $P < 0.05$; **, $P < 0.01$; ***, $P < 0.001$; ns, $P > 0.05$).

particular when they produce reactive oxygen species (ROS). We used Mitosox staining to quantify ROS produced under altered mitochondria function, but this assay did not reveal any difference between Atg5$^{ΔCd207}$ and Atg5$^{WT}$ mice (Fig. 6 K). All things considered, since mitochondria and lysosomes of autophagy-deficient LCs remained functional, we concluded that the shortage in the lipophagy-dependent fatty acid supply could not be compensated by increased mitochondrial mass.

### Excess lipid oxidation causes ferroptosis in ATG5-deficient Langerhans cells

Mitosox detects superoxide O2•−, yet other ROS may cause cellular damage and death, i.e., H$_2$O$_2$ and HO•. Interestingly, ATG5-deficient LCs showed upregulated transcription of *Gss* (Glutathione-S Synthetase), *Slc7a11* (Cysteine/glutamate antiporter xCT), *Esd* (S-formylglutathione hydrolase), and *Gclm* (Glutamate-Cysteine Ligase Modifier Subunit), which are key elements in the glutathione-dependent response to such ROS (Fig. 7 A). The glutathione pathway is notably involved in preventing ferroptosis, in which cell death occurs as a consequence of iron-dependent lipid peroxidation (Stockwell, 2022). In favor of this hypothesis, we noticed several genes that showed significant moderate or high (more than twofold) upregulation in ATG5-deficient LCs. They included ferroptosis-related genes such as *Hfe* (Homeostatic iron regulator), *Ftl1* (Ferritin Light Chain), *Sat1* (Spermidine/Spermine N1-Acetyltransferase 1), *Lpcat3* (Lysophosphatidylcholine Acyltransferase 3), and *Tfrc* (Transferrin receptor protein 1) (Fig. 7 A). Accordingly, LCs of Atg5$^{ΔCd207}$ mice had increased surface expression of CD71/Transferrin receptor, which is required for iron import into the cells (Fig. 7 B). Consistently, the lack of autophagy led LCs to accumulate ferrous iron (Fig. 7 C), which catalyzes the production of oxidated lipid species. We also noticed elevated expression of acyl-CoA synthetase long-chain family member 4 (*Acsl4*) and prostaglandin-endoperoxide synthase 2/cyclooxygenase 2 (*Ptgs2*) (Yang et al., 2014, 4), which are considered as strong indicators of ongoing ferroptosis. Finally, in RNA expression analyses based on Kyoto Encyclopedia of Genes and Genomes (KEGG) pathways, 26 out of 137 genes (19%) in the apoptosis network were differentially expressed by autophagy-deficient LCs, whereas 39% of ferroptosis-related genes (16 out of 41) were affected (Fig. S3, F and G; and Table S2).

To confirm that lipid peroxidation occurs in the absence of functional autophagy, LCs were exposed to Bodipy-C11, a derivative of undecanoic acid that emits fluorescence upon oxidation. Upon treatment with this compound, LCs harvested from Atg5$^{ΔCd207}$ mice displayed significantly higher fluorescence intensity than LCs of

Atg5$^{WT}$ mice (Fig. 7 D). Finally, we exposed enriched LCs in vitro to the specific ferroptosis inhibitor ferrostatin-1 (Dixon et al., 2012). Whereas Bodipy-C11 oxidation in Atg5$^{WT}$ LCs was unaffected, the ferrostatin-1 treatment normalized the elevated levels observed in Atg5$^{ΔCd207}$ LCs (Fig. 7 E). Altogether, autophagy-deficient LCs concomitantly show an increase in lipid peroxidation, high expression of CD71, an accumulation of Fe$^{2+}$, and sensitivity to ferrostatin-1, thereby supporting the notion that oxidation of the accumulated lipids leads them to ferroptosis, which contributes to their progressive depletion from the epidermis.

### Langerhans cells under metabolic stress have a proinflammatory signature

Besides metabolic imbalance, RNA sequencing revealed dysregulation of inflammation-related genes (Fig. S3 E and Fig. 8 A). In particular, and as confirmed by RT-qPCR (Fig. 8 B), autophagy-deficient LCs increased their expression of mRNA encoding chemokines CXCL1, CXCL2, and CXCL3, known to attract neutrophils through CXCR2. Despite this observation, no obvious signs of inflammation were observed on the skin of Atg5$^{ΔCd207}$ mice: when analyzing ear skin of 3-wk-old mice for myeloid infiltrates (Fig. 7 C), the proportions of Gr1$^+$ Ly6G$^+$ neutrophils or Gr1$^{low}$ Ly6G$^-$ monocytes did not differ between Atg5$^{WT}$ and Atg5$^{ΔCd207}$ mice in untreated conditions (Fig. 8, D–G). Since *Nlrp3*, encoding a key inflammasome component, was markedly upregulated in LCs of Atg5$^{ΔCd207}$ mice, we challenged this pathway by injecting intradermally a small dose of alum hydroxide into the ears of 3-wk-old mice. In Atg5$^{WT}$ mice, this resulted in only a modest increase of neutrophils (Fig. 8 D), whereas monocytes were not attracted (Fig. 8 E). On the other hand, alum hydroxide injection was able to drive a significant monocyte infiltration into the ears of Atg5$^{ΔCd207}$ mice (Fig. 8 E). Despite this, no difference could be demonstrated when comparing the extent of immune infiltrates of alum-treated Atg5$^{WT}$ and Atg5$^{ΔCd207}$ mice. As an alternative danger signal, we evaluated poly(I:C), which engages TLR3 and/or RIG-I signaling pathways. Poly(I:C) injection promoted the recruitment of both neutrophils and monocytes in Atg5$^{WT}$ mice (Fig. 8, F and G). Yet, inflammatory infiltrates were identical in Atg5$^{WT}$ and Atg5$^{ΔCd207}$ mice. These results suggest that autophagy-deficient LCs, although they exhibit a proinflammatory profile, do not prompt an exacerbated response to the challenges tested here.

### Autophagy deficiency affects neuronal interaction genes in Langerhans cells

Several reports showed that a long-term absence of LCs leads to a decrease in intraepidermal nerve endings, although the

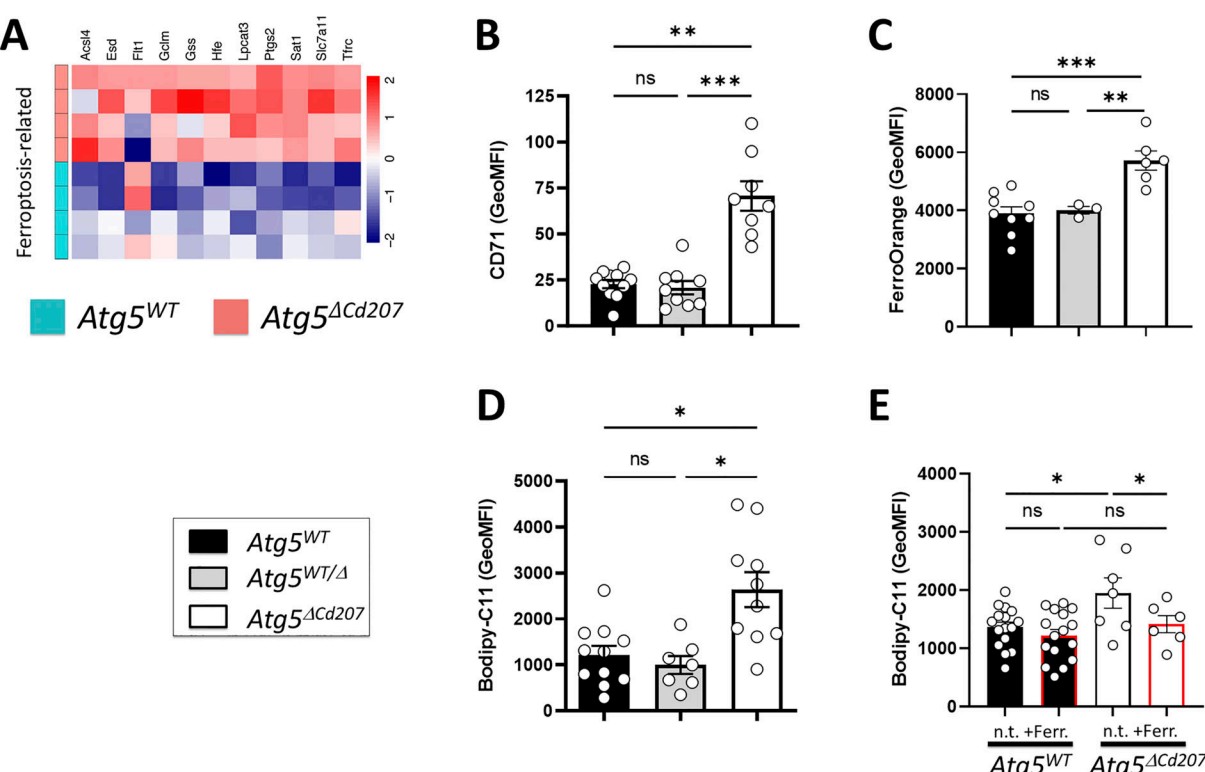

**Figure 7. Autophagy-deficient Langerhans cells undergo ferroptosis. (A)** Differentially expressed transcripts included in the KEGG ferroptosis pathway (mmu04216) in $Atg5^{WT}$ versus $Atg5^{\Delta Cd207}$ LCs. No threshold was applied to fold changes. **(B)** MFI of CD71 at the surface of LCs. **(C)** MFI of FerroOrange staining, reflecting the concentration of ferrous iron ($Fe^{2+}$) in LCs. Data are pooled from five independent experiments, with each point representing one individual 3-wk-old mouse. **(D)** Quantification of lipid peroxidation for epidermal LCs of control ($Atg5^{WT}$ and $Atg5^{WT/\Delta}$) and $Atg5^{\Delta Cd207}$ mice, as measured by MFI of Bodipy-C11. Data are pooled from 10 independent experiments, with each point representing one individual 3-wk-old mouse. **(E)** Lipid peroxidation of $Atg5^{WT}$ and $Atg5^{\Delta Cd207}$ LCs after overnight incubation with 50 µM ferrostatin-1. Data are pooled from four independent experiments, with each point representing one individual 3-wk-old mouse. Statistical analyses: (B–D) Kruskal–Wallis one-way ANOVA followed by Dunn's multiple comparison test (***, $P < 0.001$; **, $P < 0.01$; *, $P < 0.05$; ns, $P > 0.05$). **(E)** Two-way ANOVA followed by Sidak's multiple comparison tests. (*, $P < 0.05$; ns, $P > 0.05$).

pathways governing this remain unidentified (Zhang et al., 2021; Doss and Smith, 2014). In our model, autophagy-deficient LCs downregulated a set of genes involved in neuronal interactions and axonal guidance (GO:0045664, GO:0050808, GO:0071526, and GO:0030031; Fig. S3 E and Fig. 9 A). Thus, we sought to investigate the epidermal neuronal network of mice aged 6 mo and older. Since neuronal development is particularly sensitive to autophagy impairment (Hara et al., 2006), we chose to compare the density of LCs and epidermal sensory neurons of $Atg5^{\Delta Cd207}$ mice with that of $Atg5^{WT}$ and heterozygous $Atg5^{WT/-}$ mice (Fig. 9 B). As we have demonstrated for mice of this age, there is a decrease in LC density in $Atg5^{\Delta Cd207}$ epidermis but not in control mice (Fig. 9 C). In parallel, quantification of β3-tubulin staining demonstrated that only $Atg5^{\Delta Cd207}$ mice presented significantly less epidermal nerve endings as compared with $Atg5^{WT}$ mice, although a slight reduction was observed in $Atg5^{WT/-}$ mice (Fig. 9 D). Intriguingly, we found that LC numbers were correlated with the density of epidermal nerve endings and $Atg5^{WT}$ mice, but not in $Atg5^{WT/-}$ or $Atg5^{\Delta Cd207}$ mice (Fig. 9 E). This suggests that the autophagy-deficient LCs that remain in aging mice may be unable to support neuronal epidermal growth, in line with their decreased neuron-related transcripts. Thus, in accordance with previous findings, the

metabolic stress resulting from reduced autophagy in LCs may have an impact on the maintenance of the epidermal neuronal network.

## Discussion

In contrast to conventional DCs, epidermal LCs self-renew to maintain their population and are exposed in the steady state to environmental conditions (hypoxia, irradiation) that may favor autophagy. We report here a major role for autophagy in LC homeostasis. Indeed, constitutively autophagy-deficient LCs progressively disappeared from the epidermis. The depletion of autophagy-deficient LCs was not due to their emigration or decreased proliferation, suggesting cell death as the most likely explanation. Our analyses excluded a significant contribution of the ER stress response and major mitochondrial damages. In the absence of the key autophagy mediator ATG5, LCs displayed clear signs of lipid-related metabolic stress and underwent progressive depletion from the epidermis. They enhanced the production of inflammation-related transcripts and showed less expression of innervation-regulating genes, which is of great interest in the context of a decreased network of epidermal nerves when these mice were aging.

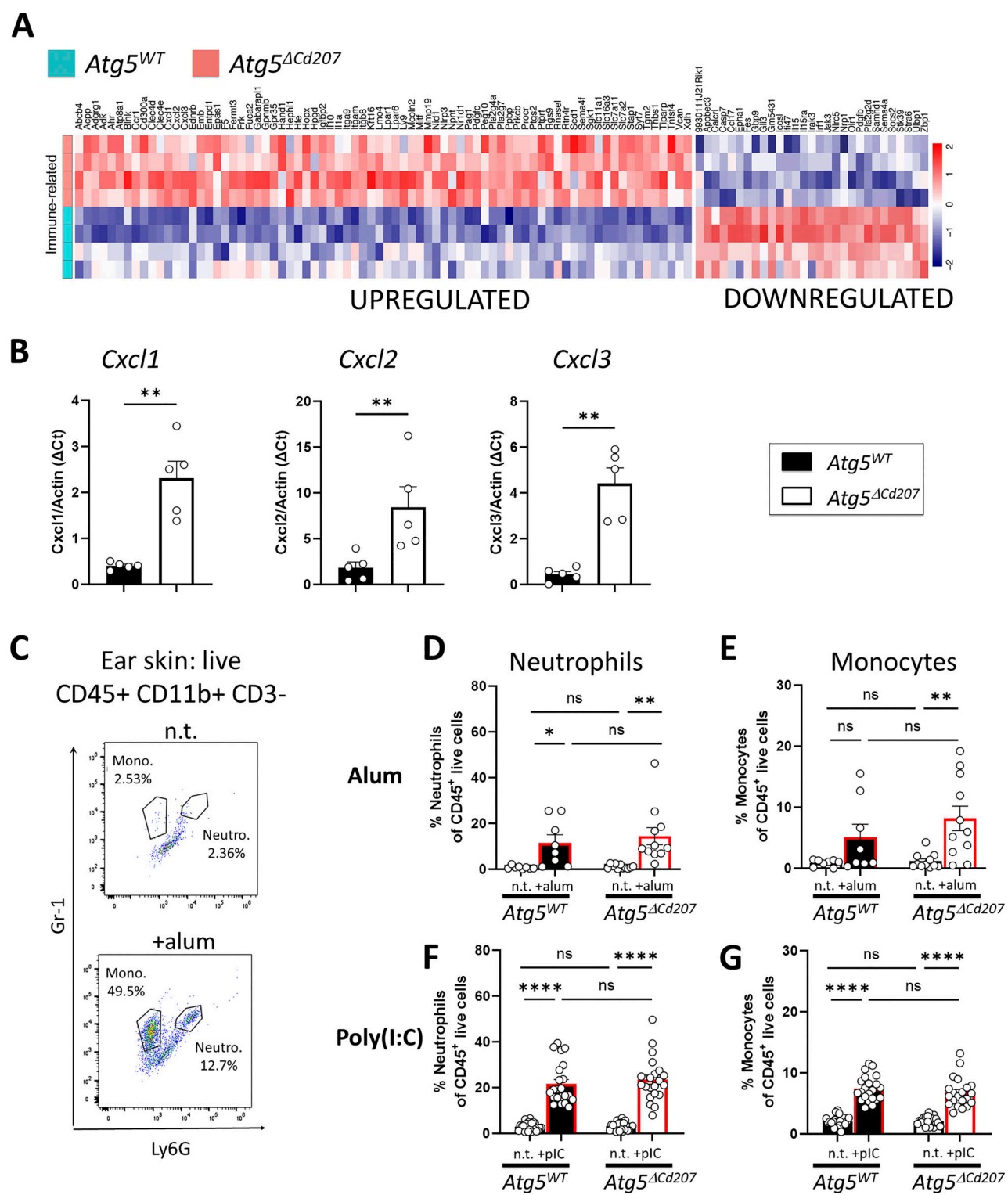

Figure 8. **Langerhans cells under metabolic stress have a proinflammatory signature. (A)** Differentially expressed transcripts related to immune function in *Atg5^WT* versus *Atg5^ΔCd207* LCs (GO:0048871, GO:0009611, GO:0001660, GO:0035458, GO:0070672, GO:0019221, GO:0097529, GO:0006954, GO:0002274, GO:0050865, GO:1905517, GO:0001773). **(B)** Expression of *Cxcl1*, *Cxcl2*, and *Cxcl3* genes quantified by RT-qPCR in purified LCs of the indicated genotypes. Data are pooled from two independent experiments, with each point representing one individual 3-wk-old mouse. **(C–G)** One ear of *Atg5^WT* or *Atg5^ΔCd207* mice was injected intradermally with 2.5 µg alum hydroxide (C–E) or 10 µg poly(I:C) (F and G) and the contralateral ear was left untreated. 4 h later, whole skin was digested and cell suspensions were monitored by flow cytometry for CD45+ CD3− CD11b+ Gr1+ Ly6G+ neutrophils and CD45+ CD3− CD11b+ Gr1 low Ly6G- monocytes (C: representative example). Percentages of neutrophils (D and F) and monocytes (E and G) were calculated as a proportion of live CD45+ cells. Data are pooled from six to nine independent experiments, with each point representing one individual 3-wk-old mouse. Statistical analysis: Mann–Whitney U test (B) or two-way ANOVA followed by Sidak's multiple comparison tests. (*, P < 0.05; **, P < 0.01; ****, P < 0.0001; ns, P > 0.05).

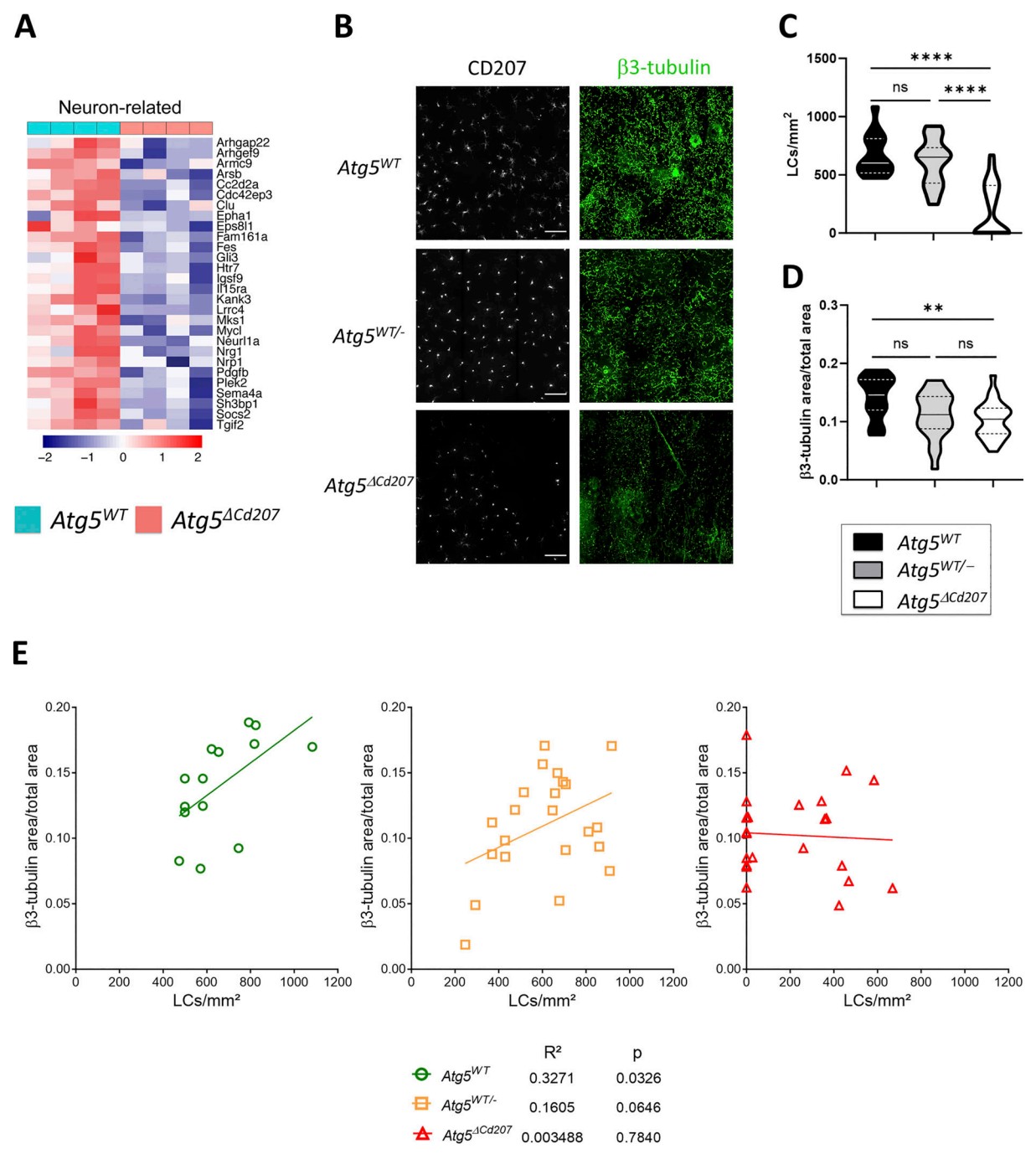

Figure 9. **Autophagy deficiency affects neuronal interaction genes in Langerhans cells. (A)** Differentially expressed genes in *Atg5*[WT] versus *Atg5*[ΔCd207] LCs of 3-wk old mice: transcripts related to neuronal interactions (GO:0030031, GO:0071526, GO:0050808, GO:0045664). **(B)** Representative immunofluorescence microscopy image of epidermal sheets obtained from ears of 6–12-mo-old *Atg5*[WT] and *Atg5*[ΔCd207] mice and stained with antibodies against β3-tubulin (neurons) and CD207 (LCs). Scale bar: 50 μm. Quantification in *Atg5*[WT], *Atg5*[WT/−], and *Atg5*[ΔCd207] mice: **(C)** Number of CD207[+] LCs per mm². **(D)** Relative area of β3-tubulin staining. Data are pooled from three independent experiments, with each point corresponding to one field of view (*n* = 3 mice per condition, at least 10 fields per mouse were scanned). Statistical analysis: Kruskal–Wallis one-way ANOVA followed by Dunn's multiple comparison test (**, P < 0.01; ****, P < 0.0001; ns, P > 0.05). **(E)** Correlation of epidermal nerve and Langerhans cell densities in 6–12-mo-old *Atg5*[WT], *Atg5*[WT/−] and *Atg5*[ΔCd207] mice. Based on epidermal sheet staining (C and D), the number of CD207[+] LCs per mm² (X axis) was plotted against the relative areas for β3-tubulin[+] nerves (Y axis). Statistical analysis: Pearson correlation test. R² correlation coefficients and P values are indicated.

The similar depletion of LCs observed using *Cd207*-specific *Atg7* deletion supports the idea that ATG5-dependent non-autophagic functions are not directly involved in the maintenance of LCs. We could also rule out lysosomal alterations or the impairment of endocytic mechanisms such as LAP since the LC network appeared unperturbed in *Rubcn*[−/−] mice. The importance of autophagy for other skin APCs in vivo has been investigated previously. An earlier report did not find any

consequence on the maintenance for CD207⁺ cDC1, in line with our findings (Lee et al., 2010), although the absence of Vps34, which plays a role in autophagy and other biological pathways, resulted in the depletion of splenic cDC1 (Parekh et al., 2017). Nevertheless, the ontogeny and features of LCs differ strikingly from those of cDC subsets (Doebel et al., 2017). Surprisingly, LCs had a continuous presence in skin-draining LNs, despite their depletion from the epidermis. This could be due to LCs remaining proliferative in older mice, which allowed the steady-state flux of LCs toward LNs to be kept constant. Emigration to LNs also entails a major environmental change as compared with the epidermis, and this may be beneficial to the extended survival of LCs.

The metabolic requirements of LCs have rarely been studied and were mostly extrapolated from those of keratinocytes nearby. In the epidermal layer where they reside, LCs are relatively distant from dermal blood capillaries and have limited access to glucose and oxygen, suggesting that fatty acids may represent a major resource. This was recently substantiated by Wculek et al., who showed that disrupting mitochondrial function led to the partial depletion of LCs (Wculek et al., 2023). In line with this, we showed a remarkable accumulation of lipid droplets when autophagy was abrogated, in addition to metabolic stress in freshly isolated LCs, revealed by AMPK phosphorylation. The seemingly unaltered ATP production could be due to normoxic in vitro culture conditions during Seahorse assays.

Interestingly, a hypoxic environment is expected to bias cellular metabolism toward an increase in lipid storage under the control of HIF-1α (Mylonis et al., 2019). We postulated that lipophagy may be important for LCs to maintain a sustainable level of lipids, limit their potential cellular toxicity, and provide free fatty acids. In this context, we propose that ferroptosis, resulting from uncontrolled supply of lipids and their peroxidation by ROS, represents one important mechanism explaining LC depletion in vivo. Ferroptosis was demonstrated by several key features of autophagy-deficient LCs: Bodipy-C11 assays, upregulation of CD71 and relevant detoxification pathways, and inhibition of lipid peroxidation by ferrostatin-1 (Stockwell, 2022). Another mechanism contributing to depletion could be apoptosis, also occurring in a proportion of ATG5-deficient LCs, most likely as a result of the critically low ATP:AMP ratio (Liang et al., 2007) revealed by increased phosphorylation of AMPK.

Epidermal hypoxia appears to regulate the functional properties of human LCs (Pierobon et al., 2016). Hypoxic tissues exhibit low levels of phosphorylated Akt (Bedogni et al., 2005), which is expected to promote autophagy. Here, LCs with disrupted autophagy displayed hyperactivation of the PI3K/Akt pathway. Although historically autophagy has been shown to be regulated through mTOR complex 1 (in particular under energetic stress), it is now clear that several induction pathways coexist with constitutive activity, especially in immune cells. Intriguingly, the progressive disappearance of epidermal LCs at a young age has been reported in mice with selective disruption of critical intermediates of the mTOR pathway (Kellersch and Brocker, 2013; Sparber et al., 2014, 2015). Therefore, since mTOR is recognized as a negative regulator of autophagy, it could be

interpreted that excessive autophagy is detrimental to LCs. However, autophagy has not been investigated in these mouse models. In addition, mTOR regulates many other cellular processes and, beyond autophagy, is critical to survival, lysosomal trafficking, or cytokine signaling pathways. By impairing the recycling of receptors through lysosomes, LAMTOR2 deletion leads to a defect in TGF-β signaling (Sparber et al., 2015), which is essential for the differentiation of LCs (Kaplan et al., 2007). Moreover, LCs deficient in Raptor, an adaptor of the mTOR complex 1, leave the epidermis (Kellersch and Brocker, 2013), which may also result from an impaired TGF-β signaling that normally maintains the epidermal LC network by restricting their spontaneous migration to skin-draining LNs (Kel et al., 2010; Bobr et al., 2012). Altogether, there is no direct evidence to date that the deleterious impact of genetic ablations affecting the mTOR pathway in LCs may depend solely on altered autophagy. Finally, the Nrf2/Keap1 pathway plays an important role in the electrophilic and oxidative stress response of cutaneous APCs, including LCs (Helou et al., 2019). In this context, it is worth noting that Keap1, which prevents the activity of transcription factor Nrf2, can be degraded through autophagy, and that Nrf2 target genes *Sqstm1* and *Hmox1* show increased expression in LCs of Atg5^ΔCd207 mice.

ATG5-deficient LCs displayed an accumulation of lipid droplets that likely resulted from impaired lipophagy. Consistent with the data reported here, a similar phenotype of lipid accumulation has been described in murine LCs in a model of imiquimod-induced psoriasis, and, strikingly, the authors also found signs of dysregulated autophagy for LCs in this model and psoriatic patients (Zhang et al., 2022b). In our selective autophagy impairment model, LCs did not promote glycolysis and were unable to take advantage of their increased respiratory capacity linked to higher mitochondrial mass, highlighting the critical importance of lipophagy for their energy production and the limitation of potentially toxic accumulation of lipids. The limited ER swelling observed in our model, which was not sufficient to trigger the UPR pathway in LCs, may be related to defective energy mobilization from lipid storage (Cubillos-Ruiz et al., 2015; Velázquez et al., 2016), multiple budding of lipid droplets (Gao et al., 2019), or modification of ER membrane dynamics following lipid peroxidation (Agmon et al., 2018). Accumulation of lipid droplets as a result of impaired autophagy machinery has been observed in other cell types that rely on lipophagy. This catabolic process is key to the development of neutrophils (Riffelmacher et al., 2017). Of note, DCs derived from *Atg5⁻/⁻* fetal liver display elevated CD36 expression and lipid droplets, although no cell death was reported during the time frame of this in vitro experiment (Oh and Lee, 2019). Yet, the lifespan of bone marrow–derived or conventional DCs is not comparable with that of LCs, in which alterations in lipid metabolism may have deleterious consequences when they accumulate over long periods of time. Considering that CD207 is expressed in LCs about 10 days after birth (Tripp et al., 2004), the time period at which LC depletion becomes visible (around 20 days of age) suggests that they cannot survive more than 2 wk to autophagy deficiency in the steady state. This represents a relatively short delay as compared with other, unrelated cell

types previously found to rely on autophagy, i.e., B-1 B cells that survive 6 wk after deletion of *Atg5* (Clarke et al., 2018).

Some features of LCs are reminiscent of tissue-resident macrophages (Doebel et al., 2017). Although autophagy regulates many functions of macrophages, it was not considered to play a prominent role in their homeostatic maintenance (Wu and Lu, 2019). The absence of autophagy affects the survival of a subset of peritoneal macrophages (Xia et al., 2020), which are of embryonic rather than monocytic origin, similar to steady-state LCs. It remains to be demonstrated whether the dependence on autophagy can be associated with the origin and/or long-term residency of macrophages within other organs.

The consequences for the epidermis of a long-term absence of LCs have been investigated through constitutive diphtheria toxin–mediated depletion in the huLangerin-DTA mouse strain (Su et al., 2020; Zhang et al., 2021). However, since LCs are absent at birth, it is difficult to identify which of the genes that they normally express may affect epidermal homeostasis. Here, we were able to document transcriptome alterations of ATG5-deficient LCs that are still present in young mice, albeit in a metabolically stressed state. First and foremost, our data suggested a potential for supporting inflammation. Yet, cutaneous immune infiltration did not occur spontaneously, and inflammatory challenges by alum hydroxide or poly(I:C) did not result in a larger immune infiltration in mice where LCs were impaired for autophagy. This may be explained by the fact that autophagy-deficient LCs progressively disappear, limiting their capacity to induce inflammation. Second, several genes involved in neuronal interactions were downregulated by ATG5-deficient LCs, including EPH receptor A1, Semaphorin 4A, Neuropilin-1, and Neuregulin-1. We investigated aging mice (6 mo and older), based on the report that a significant decrease in epidermal nerve endings requires LCs to be completely absent for 30 days (Langerin-DTR model) (Zhang et al., 2021). Similarly, the epidermal nerve network was impaired in *Atg5*$^{\Delta Cd207}$ mice in which the density of LCs had been severely reduced. However, a significant impairment of the neuronal network did not occur before at least 3 mo of age, possibly because *Atg5*$^{\Delta Cd207}$ mice retain enough LCs to support nerve growth. Therefore, it was not possible to draw further conclusions on epidermal nerve maintenance by autophagy-deficient LCs. Despite this, our findings may represent a milestone for future investigations on neuro-immune interactions, considering the putative role of LCs and dermal macrophages (Kolter et al., 2019) in regulating sensory neuron growth and repair in the skin.

Altogether, we shed light on the metabolic adaptations of LCs that ensure their long-lasting tissue residency. It will be of great interest to translate these findings in the context of human skin diseases, considering that the lipid supply and autophagy capacity of LCs may perturb their homeostasis and favor inflammation.

## Materials and methods
### Mice
Mice were bred and maintained on a C57BL/6J background at the animal facility of the Institut de Biologie Moléculaire et Cellulaire (IBMC). *Atg5*$^{flox/flox}$ and *Cd207-cre* mice were gifted by N. Mizushima and B.E. Clausen, respectively (Hara et al., 2006; Zahner et al., 2011). *Atg5*$^{+/-}$ mice were generated at the IBMC (Arnold et al., 2016). [*Atg5*$^{+/-}$; *Cd207-cre*] were obtained from a first cross between *Cd207-cre* and *Atg5*$^{+/-}$, then bred to *Atg5*$^{flox/flox}$ to obtain [*Atg5*$^{flox/-}$; *Cd207-cre*] (*Atg5*$^{\Delta Cd207}$) and littermates [*Atg5*$^{flox/+}$; *Cd207-cre*] (*Atg5*$^{WT/\Delta}$) and [*Atg5*$^{flox/+}$] (*Atg5*$^{WT}$). Mice were genotyped for their *Atg5* allele and the *Cd207-cre* transgene as previously described (Arnold et al., 2016; Zahner et al., 2011). All mice were bred and maintained in accordance with the guidelines of the local institutional Animal Care and Use Committee (CREMEAS).

### Cell preparation and culture
*Lymph nodes*
Brachial and inguinal lymph nodes were digested for 1 h at 37°C under shaking in R2 buffer (RPMI-1640 medium containing L-glutamine [Lonza] plus 2% fetal calf serum [Dutscher]) supplemented with 50 µg/ml DNAse and 10 µg/ml collagenase D (Roche).

*Digestion of back skin epidermis (electron microscopy, LC proportions, caspase-3 activation, proliferation assays)*
Back skin was incubated with 0.25% Trypsin (VWR) for 45 min at 37°C. After removal of the dermis, the epidermis was teased apart with forceps, followed by 15 min of gentle shaking on a rotating wheel. Where indicated, CD11b$^+$ LCs were enriched by magnetic bead separation (Miltenyi-Biotec).

*Ear skin digestion (skin DC subsets, quantification of immune infiltrates)*
Ear skin was cut into small pieces, digested in R2 buffer containing 0.15 mg/ml LiberaseTM and 0.12 mg/ml DNAse (Roche) for 45 min at 37°C, and filtered through 100-µm cell strainers.

*Epidermal crawl-out assay (Bodipy C16 and glucose uptake, in vitro treatment with inhibitors, Seahorse assay)*
Back skin was incubated overnight at 4°C in R2 buffer, containing 1 mg/ml dispase II (Roche). The separated epidermis was then laid upon cell culture medium (RPMI-1640 medium supplemented with 10% fetal calf serum, 50 mM β-Mercaptoethanol (Gibco), 1% Gentamicin (Gibco), and 10 mM HEPES (Lonza) in a Petri dish for 24 h at 37°C, allowing emigration of LCs.

*Bone marrow-derived DCs*
Femurs and tibias were collected from C57BL/6J mice. Bone marrow was flushed out, red blood cells were lysed, filtered, and cultured for 7 days in complete RPMI medium (RPMI-1640 medium containing L-glutamine plus 10% fetal calf serum) containing 20 ng/ml recombinant GM-CSF (Peprotech).

### Electron microscopy
Epidermal cell suspensions (freshly isolated or cultured for 3 days) were processed for electron microscopy essentially as described (Cavinato et al., 2017). Briefly, after pre-enrichment on bovine serum albumin density gradient, cells were washed and fixed using Karnovsky's formaldehyde–glutaraldehyde

fixative for 1 h at room temperature. Specimens were post-fixed in aqueous 3% osmium tetroxide and contrasted with 0.5% veronal-buffered uranyl acetate. Dehydration of samples was done in a graded series of ethanol concentrations, followed by embedding in Epon 812 resin. Ultrathin sections were mounted on nickel grids, contrasted with lead citrate, and examined by transmission electron microscopy (Phillips EM 400; Fei Company Electron Optics) at an operating voltage of 80 kV. LCs were identified within epidermal cell suspensions by their electron-lucent cytoplasm, the absence of keratin tonofilament bundles, the presence of cytoplasmic processes (dendrites), and their ultrastructural hallmarks, the Birbeck granules.

### Antibodies and reagents for flow cytometry and immunofluorescence microscopy
Antibody staining for flow cytometry or immunofluorescent microscopy was performed in SE buffer (fetal calf serum 2%, EDTA 2.5 mM). All reagents and antibodies are listed in Table 1.

### Autophagy flux assessment by flow cytometry
Measurements of autophagy fluxes were carried out using the Guava Autophagy LC3 Antibody-based Detection Kit (Luminex). Immature LCs, isolated by anti-CD11b magnetic bead separation (Miltenyi-Biotec) from fresh epidermal suspensions, or mature LCs, enriched by epidermal crawl-out cultures, were cultured 18 h at 37°C with or without the lysosome inhibitor provided with the kit (60 µM hydroxychloroquine). After labeling by FVD450, cells were stained for CD45, I-A/I-E, and TCRγ/δ. Cells were permeabilized with 0.05% saponin (Merck Millipore) to wash out the cytosolic LC3-I, then membrane-associated LC3 (LC3-II) was preferentially stained with anti-LC3 FITC (clone 4E12). Flow cytometry analysis allowed for the calculation of autophagy fluxes, dividing the LC3-FITC mean fluorescence intensities (MFI) of treated cells by that of untreated cells.

### Phosflow staining
Measurement of the phosphorylation levels of proteins Akt, 4E-BP1 and S6 was carried out using Phosflow antibodies and buffers (BD Bioscience). Briefly, 500,000 epidermal crawl-out cells were fixed by the incubation in 1 ml of pre-warmed 1X Phosflow Fix Buffer I for 30 min at 37°C. After washing, cells were stained for CD45, MHCII (I-A/I-E), and CD3ε, then permeabilized by incubating cells in 1 ml of cold Phosflow Perm Buffer III for 30 min on ice. Finally, cells were stained by Phosflow antibodies diluted 1/20 in 100 µl vol of SE buffer and left at room temperature for 30 min in the dark. Samples were analyzed on an Attune NxT flow cytometer (Invitrogen) to measure the geometric MFI of p-Akt, p-4E-BP1 and p-S6 in CD45$^+$ CD3ε– MHCII+ LCs, CD45$^+$ CD3ε+ MHCII- epidermal T cells and CD45– keratinocytes.

### Glucose uptake
Cells obtained by crawl-out were glucose-starved for 24 h in PBS (Lonza) supplemented with 0.5% fetal calf serum for 8 h. Cells were then incubated for 30 min at 37°C with 150 µM of 2-[N-(7-nitrobenz-2-oxa-1,3-diazol-4-yl) amino]-2-deoxy-D-glucose (Thermo Fisher Scientific).

### Pharmacological inhibitions
Cells obtained by crawl-out were incubated for 24 h at 37°C with the phosphatidylinositol-3-kinase inhibitor wortmannin or the carnitine palmitoyltransferase-1 inhibitor etomoxir (both from Sigma-Aldrich), at 10 and 200 µM respectively.

5-bromo-2′-deoxyuridine incorporation 1 mg of 5-bromo-2′-deoxyuridine (BrdU; Sigma-Aldrich) was administered by intraperitoneal injection 72 h prior to analysis. Drinking water also contained 0.8 mg/ml BrdU. Following staining of surface markers CD45, I-A/I-E and TCRγδ, epidermal single-cell suspensions were fixed with Cytofix/Cytoperm buffer (BD Biosciences) and permeabilized with permeabilization buffer (BD Biosciences). DNA was then denatured with a DNAse solution (100 µg/ml; BD Biosciences) to improve the accessibility of the incorporated BrdU to the detection antibody.

### Quantitative real-time RT-PCR analysis
RNA was extracted from cells sorted from lymph nodes or epidermis on a FACS Melody cell sorter (BD Biosciences) with RNeasy microKit (Qiagen) for lymph nodes and Trizol for epidermis (Thermo Fisher Scientific). cDNA was obtained with Maxima Reverse Transcriptase Kit (Thermo Fisher Scientific) using a T100 Thermal cycler (Biorad) for *Atg5* quantification and with RevertAid H Minus First Strand cDNA Synthesis Kit (Thermo Fisher Scientific) for *Cxcl1*, *Cxcl2*, and *Cxcl3* quantification. Quantitative real-time PCR was performed on cDNA using Taqman preAmp MasterMix and Taqman Universal Mastermix (Thermo Fisher Scientific) and Assays-on-Demand probes (*Gapdh*: Mm03302249_g1; *Atg5*: Mm00504340_m1) for *Atg5* quantification and Fast SYBR Green Master Mix (Thermo Fisher Scientific) with primers Actb (β-Actin) forward: 5′-CAT TGCTGACAGGATGCAGAAGG-3′; β-*Actin* reverse: 5′-TGCTGG AAGGTGGACAGTGAGG-3′; *Cxcl1* forward: 5′-TCCAGAGCTTGA AGGTGTTGCC-3′; *Cxcl1* reverse: 5′-AACCAAGGGAGCTTCAGG GTCA-3′; *Cxcl2* forward: 5′-CATCCAGAGCTTGAGTGTGACG-3′; *Cxcl2* reverse:5′-GGCTTCAGGGTCAAGGCAAACT-3′; *Cxcl3* forward: 5′-TGAGACCATCCAGAGCTTGACG-3′; *Cxcl3* reverse: 5′-CCTTGGGGGGTTGAGGCAAACTT-3′ (Eurogentec) for the quantification of *Cxcl1*, *Cxcl2* and *Cxcl3*. Each sample was amplified in triplicate or duplicate in a StepOnePlus real-time PCR system (Applied Biosystems). mRNA levels were calculated with the StepOne v2.1 software (Applied Biosystems), using the comparative cycle threshold method, and normalized to the mean expression of *Gapdh* (for *Atg5*) and *Actb* (for *Cxcl1*, *Cxcl2* and *Cxcl3*) housekeeping genes.

### Immunofluorescence microscopy of epidermal sheets
Ear epidermis was separated from the dermis by ammonium thiocyanate digestion (0.15 M) for 20 min at 37°C. Alternatively, for optimal preservation of neuronal networks, epidermal sheets were separated after 10 mM EDTA diluted in PBS for 1 h. Epidermis was then fixed by incubation in PBS 4% PFA or in glacial acetone for 15 min at 4°C followed by incubation with PBS 5% BSA 0.1% Triton. Primary antibodies were incubated overnight at 4°C. After fixation, epidermal sheets were washed four times in blocking buffer consisting in 5% BSA in PBS for 15 min each time at room temperature. Sheets were then incubated

Table 1.   **Antibodies and reagents for flow cytometry and immunofluorescence microscopy**

| Antibody target or reagent | Fluorochrome | Clone | Supplier | Reference |
|---|---|---|---|---|
| Perilipin-2/ADFP | Uncoupled | EPR3713 | Abcam | ab108323 |
| Phospho-AMPKα (T183/T172) | Uncoupled | Polyclonal | Abcam | ab23875 |
| SQSTM1/p62 | Uncoupled | 2C11 | Abcam | ab56416 |
| Active Caspase-3 | FITC | C92-605 | BD Biosciences | 559341 |
| BrdU | FITC | B44 | BD Biosciences | 552598 |
| CD103 | PE | M290 | BD Biosciences | 557495 |
| CD11c | PerCP-Cy5.5 | HL3 | BD Biosciences | 560584 |
| CD36 | PE | CRF D-2712 | BD Biosciences | 562702 |
| CD3ε | FITC | 145-2C11 | BD Biosciences | 553062 |
| CD71 | FITC | C2 | BD Biosciences | 553266 |
| CD8α | APC | 53-6.7 | BD Biosciences | 561093 |
| Gr-1 | PE | RB6-8C5 | BD Biosciences | 553128 |
| I-A/I-E | Biotinylated | M5114.15.2 | BD Biosciences | 553622 |
| Ki67 | PerCP-Cy5.5 | B56 | BD Biosciences | 561284 |
| Ly-6G | FITC | 1A8 | BD Biosciences | 551460 |
| β3-tubulin | Uncoupled | TUJ1 | Biolegend | 801202 |
| CD11b | PerCP-Cy5.5 | M1/70 | Biolegend | 101228 |
| CD3ε | APC | 145-2C11 | Biolegend | 100312 |
| CD3ε | PerCP-Cy5.5 | 145-2C11 | Biolegend | 100328 |
| CD45 | PE-Cy7 | 30F11 | Biolegend | 103114 |
| CD45 | APC-Cy7 | 30F11 | Biolegend | 103116 |
| CD86 | PE | GL-1 | Biolegend | 105008 |
| I-A/I-E | AlexaFluor 700 | M5114.15.2 | Biolegend | 107622 |
| TCRγδd | PE | GL3 | Biolegend | 118107 |
| Fixable viability dye | eFluor450 | N/A | eBioscience | 65-0863-14 |
| Fixable viability dye | eFluor780 | N/A | eBioscience | 65-0865-14 |
| I-A/I-E | PE | M5114.15.1 | eBioscience | 12-5321-81 |
| TCRγδ | APC | GL3 | eBioscience | 17-5711-82 |
| CD207 | AlexaFluor 647 | 929F3 | Eurobio/Dendritics | DDX0362A647 |
| CD207 | AlexaFluor 488 | 929F3 | Eurobio/Dendritics | DDX0362A488 |
| Bodipy | Bodipy 493/503 | N/A | Invitrogen | D3922 |
| Bodipy C11 | Bodipy 581/591 | N/A | Invitrogen | D3861 |
| Bodipy FL C16 | Bodipy 505/512 | N/A | Invitrogen | D3821 |
| DAPI | N/A | N/A | Invitrogen | D3571 |
| Donkey anti Rabbit IgG | AlexaFluor 647 | Polyclonal | Invitrogen | A31573 |
| ER tracker Blue/White DPX | N/A | N/A | Invitrogen | E12353 |
| Lysosensor green DND-189 | N/A | N/A | Invitrogen | L7535 |
| Lysotracker red DND-99 | N/A | N/A | Invitrogen | L7528 |
| Mitosox red | N/A | N/A | Invitrogen | M36008 |
| Mitotracker deep red 633 | N/A | N/A | Invitrogen | M22426 |
| Mitotracker green FM | N/A | N/A | Invitrogen | M7514 |
| Mouse IgG(H+L) | AlexaFluor 555 | Polyclonal | Invitrogen | A31570 |
| Mouse IgG(H+L) | AlexaFluor 594 | Polyclonal | Invitrogen | A11032 |
| Streptavidin | AlexaFluor 488 | N/A | Invitrogen | S11223 |

Table 1. **Antibodies and reagents for flow cytometry and immunofluorescence microscopy (Continued)**

| Antibody target or reagent | Fluorochrome | Clone | Supplier | Reference |
|---|---|---|---|---|
| Streptavidin | AlexaFluor 546 | N/A | Invitrogen | S11225 |
| Poly (I:C) | N/A | N/A | InvivoGen | tlrl-pic |
| Guava autophagy LC3 assay kit | FITC | 4E12 | Luminex | FCCH100171 |
| CD11b-coupled Microbeads | N/A | N/A | Miltenyi Biotec | 130-049-601 |
| 2-NBDG | N/A | N/A | Sigma-Aldrich | 72987 |
| Aluminium hydroxide | N/A | N/A | Sigma-Aldrich | 239186 |
| Etomoxir | N/A | N/A | Sigma-Aldrich | E1905 |
| Ferrostatin-1 | N/A | N/A | Sigma-Aldrich | SML0583 |
| Wortmannin | N/A | N/A | Sigma-Aldrich | W1628 |
| Fast SYBR green master mix | N/A | N/A | Thermo Fisher Scientific | 4385612 |
| RevertAid H minus first strand cDNA synthesis kit | N/A | N/A | Thermo Fisher Scientific | K1632 |
| GLUT1 | Uncoupled | EPR3915 | Abcam | ab115730 |
| BioTracker FerroOrange | N/A | N/A | Sigma-Aldrich | SCT210 |
| BD Phosflow fix buffer I | N/A | N/A | BD Biosciences | 557870 |
| BD Phosflow Perm buffer III | N/A | N/A | BD Biosciences | 558050 |
| Phospho-4E-BP1 (pT36/pT45) | AlexaFluor 488 | M31-16 | BD Biosciences | 560287 |
| Phospho-S6 (pS244) | PE | N5-676 | BD Biosciences | 560462 |
| Phospho-Akt (pS473) | Brilliant Violet 421 | M89-61 | BD Biosciences | 562599 |

overnight at 4°C with the primary antibodies: anti-β3-tubulin and AF647 anti-CD207 diluted in blocking buffer. After washing the sheets as described above, they were incubated with a solution of goat anti-mouse AF594, and 4′,6-diamidino-2-phenyl-indole (DAPI) in a blocking buffer for 1 h at room temperature. After additional washings, epidermal sheets were mounted in Fluoromount-G mounting medium (Thermo Fisher Scientific) and observed under a confocal microscope (Yokogawa Spinning Disk; Zeiss). Whole-mount epidermal images were processed using the open-source software FIJI to measure the total analyzed area for each sample and to quantify the MFI.

### Immunofluorescence microscopy of epidermal cell suspensions

Cell suspensions were deposited on Lab-Tek chamber slides (Thermo Fisher Scientific Nunc) previously coated with a poly-L-Lysine solution (Sigma-Aldrich) diluted in ultrapure water at 0.02% (vol/vol) to enhance cellular adhesion. Epidermal cells were then incubated with Mitotracker, Mitosox, ER-tracker, Bodipy, or Bodipy-C16 according to the manufacturer's instructions (Invitrogen) before fixation using 2% PFA in PBS for 15 min at RT. DAPI was incubated for 15 min at RT. Tissues were mounted and observed under a confocal microscope (Yokogawa Spinning Disk, Zeiss).

### RNA sequencing

Total RNA was isolated from $10^5$ sorted LCs with the RNeasy Mini Kit (Qiagen). RNA integrity was evaluated on an Agilent Bioanalyzer 2100 (Agilent Technologies). Total RNA Sequencing libraries were prepared with SMARTer Stranded Total RNA-Seq Kit v2 - Pico Input Mammalian (TaKaRa) according to the manufacturer's protocol. Briefly, random priming was used for first-strand synthesis and ribosomal cDNA was cleaved by ZapR v2 in the presence of mammalian R-probes V2. Libraries were pooled and sequenced (paired-end 2*75 bp) on a NextSeq500 using the NextSeq 500/550 High Output Kit v2 according to the manufacturer's instructions (Illumina). For analysis, quality control of each sample was carried out and assessed with the NGS Core Tools FastQC (https://www.bioinformatics.babraham. ac.uk/projects/fastqc/). Sequence reads were mapped on the GRCm38 reference genome using STAR (Dobin et al., 2013) and unmapped reads were remapped with Bowtie2 (Langmead and Salzberg, 2012) using a very sensitive local option to optimize the alignment. The total mapped reads were finally available in BAM (Binary Alignment Map) format for raw read count extraction. Read counts were found by the HTseq-count tool of the Python package HTSeq (Anders et al., 2015) with default parameters to generate an abundance matrix. Finally, differential analyses were performed using the DEseq2 (Love et al., 2014) package of the Bioconductor framework. Differentially expressed genes between $Atg5^{\Delta Cd207}$ and $Atg5^{WT}$ were selected based on the combination of adjusted P value <0.05 and FDR < 0.1, with fold changes less than –2 or >2, unless otherwise stated in figure legends. Pathway enrichment analysis was performed using Metascape (https://metascape.org) (Zhou et al., 2019).

Finally, to highlight the genes that are part of the ferroptosis or the apoptosis pathways, we screened the differentially expressed genes (P < 0.05) for their presence in the entries of the respective KEGG pathways using Excel (Table S2), with https://www.genome.jp/pathway/mmu04216 for ferroptosis and

https://www.genome.jp/entry/mmu04210 for apoptosis. The genes that overlapped with the pathway genes were then labeled using the visualization tool of KEGG.

## Metabolic parameter quantitation by extracellular flux assay

CD45$^+$ MHCII$^+$ CD207$^+$ CD103$^-$ TCR$\gamma\delta^-$ LCs were sorted from epidermal crawl-out suspensions on a FACSFusion cell sorter (Becton-Dickinson). Purified LCs or BMDCs ($2.10^5$ cells/well) were seeded in Seahorse XF96 culture plate coated with poly lysine (Sigma-Aldrich). After overnight culture, a Mitochondrial Stress Test was performed. In this assay, culture wells are injected sequentially with different inhibitors of mitochondrial respiration. Energy production resulting from mitochondrial respiration was determined after each injection by measuring oxygen consumption rates (OCR, pmoles/min) on a Seahorse XF96 according to the manufacturer's instructions (Agilent). Oligomycin (OM) injection allowed for calculating the oxygen consumption used for mitochondrial ATP synthesis. Carbonyl cyanide 4-(trifluoromethoxy) phenylhydrazone (FCCP) uncoupled mitochondrial respiration, allowing for the calculation of maximal respiration and spare respiratory capacity. Finally, rotenone (ROT) and antimycin A (AA) blocked mitochondrial complexes I and III to determine the non-mitochondrial oxygen consumption. The following metabolic parameters were calculated:

$$ATP\ production = OCR_{baseline} - OCR_{OM}$$
$$Maximum\ respiration = OCR_{FCCP} - OCR_{AA+ROT}$$
$$Spare\ respiratory\ capacity (SRC) = OCR_{FCCP} - OCR_{baseline}.$$

## Lipid peroxidation assay

50,000 enriched CD11b+ LCs were seeded and incubated for 10 min at 37°C with 2 mM Bodipy-C11 (581/591) (4,4-difluoro-5,7-dimethyl-4-bora-3a,4a-diaza-s-indacene-3-undecanoic acid; Invitrogen) in PBS. Cells were then resuspended in SE buffer and incubated with the following antibodies: CD3ε-PerCP-Cy5.5, MHCII-AF700, and CD45-APC-Cy7. Upon gating on CD45$^+$ CD3$^-$ MHCII$^+$ cells, the fluorescence of Bodipy-C11 was collected from the FITC channel on a Gallios cytometer (Beckman-Coulter).

## In vitro ferroptosis inhibition

At least 50,000 enriched CD11b+ LCs were seeded and incubated overnight in a complete RPMI medium and 50 μM ferrostatin-1 or DMSO (untreated control). Cells were then stained and analyzed as indicated for the lipid peroxidation assay.

## Detection of intracellular Fe$^{2+}$ with FerroOrange

To detect intracellular iron levels, 100,000 enriched CD11b$^+$ LCs were incubated with 1 μM FerroOrange (Sigma-Aldrich) in PBS for 30 min at 37°C. After incubation, the cells were washed in PBS and stained for flow cytometry analysis. Upon gating on CD45$^+$ CD3$^-$ MHCII$^+$ cells, the fluorescence of FerroOrange was collected from the PE channel on a Gallios cytometer (Beckman-Coulter).

## Induction of cutaneous inflammation

For each mouse, one ear was injected intradermally with 25 μl of 100 μg/ml alum hydroxide (Roche), and the contralateral ear was left untreated. 4 h later, whole skin was digested and cell suspensions were monitored by flow cytometry for CD45$^+$ CD3$^-$ CD11b$^+$ Gr1$^+$ Ly6G$^+$ neutrophils and CD45$^+$ CD3$^-$ CD11b$^+$ Gr1$^+$ Ly6G$^-$ monocytes.

## Quantification of the density of epidermal nerve endings

The open-source software iLastik was used to segment images of whole-mount epidermal sheets stained for β3-tubulin and CD207, using machine learning to differentiate the background from β3 tubulin signal. Images were then processed using the open-source software FIJI to measure the total area of each scan, as well as the area that was determined positive for β3-tubulin.

## Statistical analyses

Statistical significance was calculated with the indicated tests using Prism software (GraphPad, versions 6-9). All data were presented as mean ± standard error of the mean (SEM). P values <0.05 were considered statistically significant (*, $P < 0.05$; **, $P < 0.01$; ***, $P < 0.001$; ****, $P < 0.0001$).

## Online supplemental material

Fig. S1 shows autophagosomes are detectable in murine Langerhans cells. Fig. S2 shows efficient deletion of *Atg5* in CD207+ DC subsets affects Langerhans cells but not cDC1. Fig. S3 shows lack of autophagy alters the transcriptome of Langerhans cells. Table S1 shows differentially expressed genes between Langerhans cells of *Atg5$^{WT}$* and *Atg5$^{\Delta Cd207}$* mice. Table S2 shows analysis of the ferroptosis and apoptosis KEGG pathways for dysregulated genes in Langerhans cells of *Atg5$^{WT}$* versus *Atg5$^{\Delta Cd207}$* mice. Video 1 shows autophagosome staining of *Atg5$^{WT}$* Langerhans cells. Video 2 shows autophagosome staining of *Atg5$^{\Delta Cd207}$* Langerhans cells. Video 3 shows endoplasmic reticulum staining of *Atg5$^{WT}$* Langerhans cells. Video 4 shows endoplasmic reticulum staining of *Atg5$^{\Delta Cd207}$* Langerhans cells. Video 5 shows lipid droplets of *Atg5$^{WT}$* Langerhans cells. Video 6 shows lipid droplets of *Atg5$^{\Delta Cd207}$* Langerhans cells.

## Data availability

Data are available in the article itself and its supplementary materials.

## Acknowledgments

We thank Pr. Noboru Mizushima for his gift of *Atg5$^{flox/flox}$* mice, Delphine Lamon and Fabien Lhericel for mouse breeding at the IBMC animal facility, as well as Claudine Ebel and Muriel Philipps at the cell sorting facility of the IGBMC.

This work was funded by the French Centre National de la Recherche Scientifique, the Laboratory of Excellence Medalis (ANR-10-LABX-0034), EquipEx program I2MC (ANR-11-EQPX-022), Strasbourg University, Fondation Arthritis Courtin, ANR ERAPerMed BATMAN (ANR-18-PERM-0001), ANR AUTOMATE (ANR-20-CE15-0018-01), ANR AURIGENE (ANR-20-CE93-0001) and Strasbourg's Interdisciplinary Thematic Institute (ITI) for Precision Medicine, TRANSPLANTEX NG, as part of the ITI 2021-2028 program of the University of Strasbourg, Centre National de la Recherche Scientifique and Institut National de la

Santé et de la Recherche Médicale, funded by IdEx Unistra (ANR-10-IDEX-0002) and SFRI-STRAT'US (ANR-20-SFRI-0012). Florent Arbogast, Delphine Bouis, and Quentin Frenger were recipients of pre-doctoral fellowships from the Ministère de la Recherche et de l'Enseignement Supérieur. Benjamin Voisin was supported by Marie Slodowska-Curie Individual Fellowship (H2020-MSCA-IF-2019 896095 VirIVITES).

Author contributions: F. Arbogast: Conceptualization, Data curation, Formal analysis, Investigation, Validation, Writing - original draft, Writing - review & editing, R. Sal-Carro: Formal analysis, Investigation, Methodology, Visualization, Writing - review & editing, W. Boufenghour: Data curation, Formal analysis, Investigation, Validation, Q. Frenger: Investigation, D. Bouis: Investigation, Writing - original draft, L. Filippi De La Palavesa: Investigation, J.-D. Fauny: Methodology, O. Griso: Methodology, Resources, H. Puccio: Resources, R. Fima: Formal analysis, Investigation, Resources, Validation, T. Huby: Resources, E.L. Gautier: Funding acquisition, Resources, Writing - review & editing, A. Molitor: Investigation, R. Carapito: Data curation, Formal analysis, Investigation, S. Bahram: Funding acquisition, Resources, N. Romani: Resources, B.E. Clausen: Resources, Writing - review & editing, B. Voisin: Conceptualization, Formal analysis, C.G. Mueller: Funding acquisition, Project administration, Resources, Writing - review & editing, F. Gros: Conceptualization, Data curation, Formal analysis, Funding acquisition, Investigation, Methodology, Project administration, Resources, Supervision, Validation, Visualization, Writing - original draft, Writing - review & editing, V. Flacher: Conceptualization, Data curation, Formal analysis, Funding acquisition, Investigation, Methodology, Project administration, Supervision, Validation, Visualization, Writing - original draft, Writing - review & editing.

Disclosures: The authors declare no competing interests exist.

Submitted: 27 March 2024

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

# Supplemental material

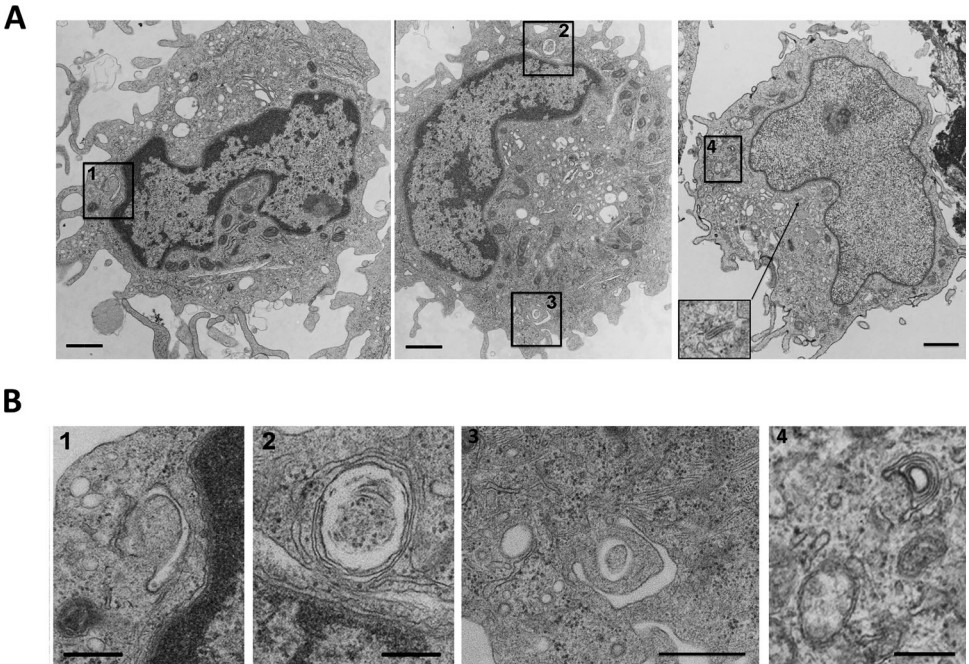

Figure S1. **Autophagosomes are detectable in murine Langerhans cells. (A)** Transmission electron microscopy of LCs in a bulk epidermal cell suspension from C57BL/6 mice, either freshly isolated (right panel) or cultured for 3 days (left and center panels). The inset image in the right panel highlights a Birbeck granule (arrow). **(B)** Close-up micrographs of autophagic structures corresponding to the boxes in the low-power overview micrographs. 1 and 3 appear to be limiting membranes of incipient autophagy; 2 and 4 show double membrane-limited autophagosomes. Scale bars: 1 µm (A); 500 nm (B).

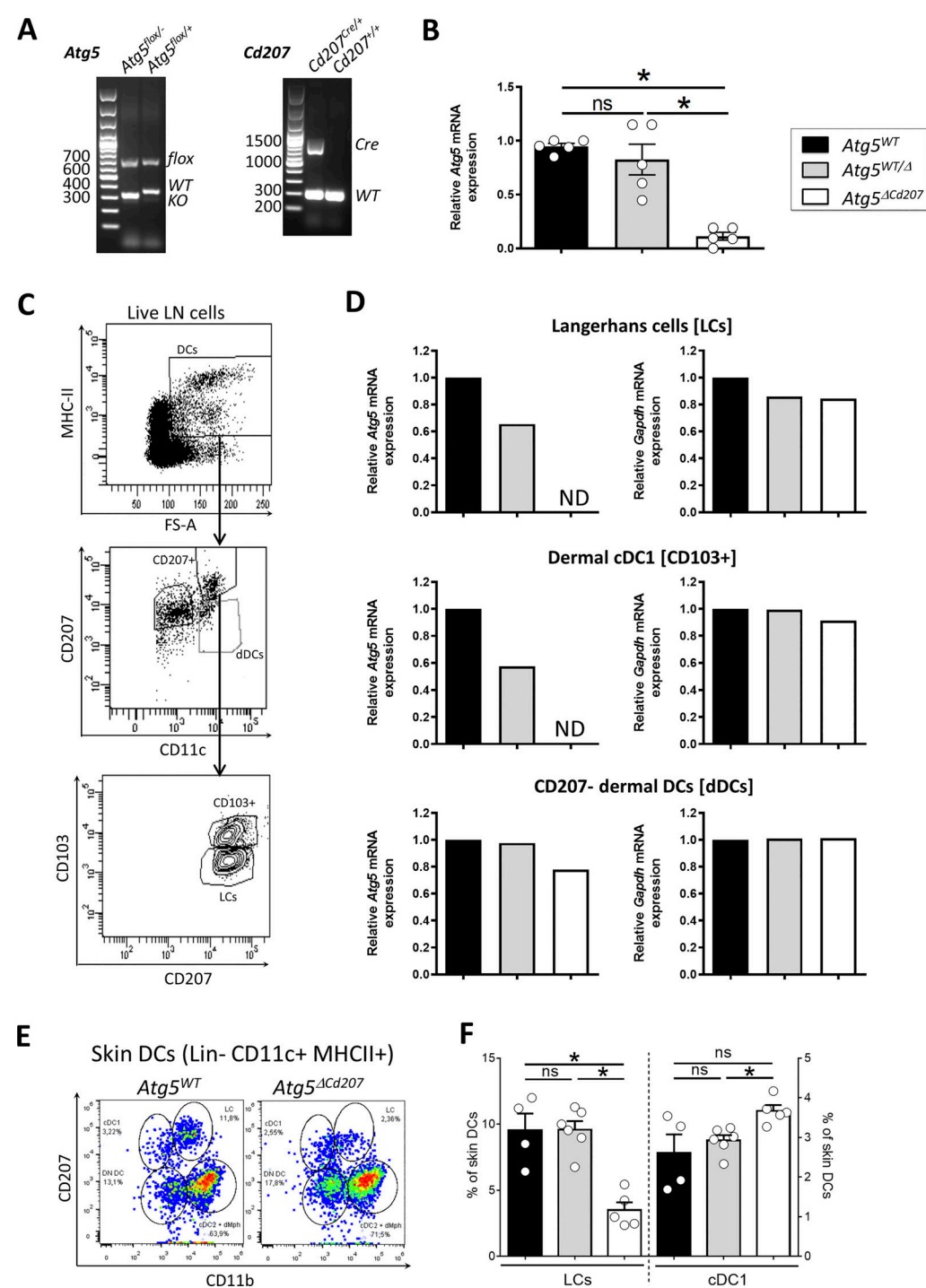

Figure S2. **Efficient deletion of *Atg5* in CD207+ DC subsets affects Langerhans cells but not cDC1. (A)** Representative electrophoresis of genotyping PCR. Left panel: floxed, wild-type (WT) and exon 3-deleted (KO) alleles of *Atg5*. Right panel: wild-type and *Cre* knock-in alleles of *Cd207*. **(B)** *Atg5* mRNA expression in sorted epidermal CD45+ MHCII+ TCRγδ− LCs from control (*Atg5^WT^* and *Atg5^WT/Δ^*) and *Atg5^ΔCd207^* mice. Fold changes were calculated relative to mRNA expression in LCs of *Atg5^WT^* control mice. Data are pooled from at least three independent experiments, with each point corresponding to one individual mouse. Statistical analysis: Kruskal–Wallis one-way ANOVA followed by Dunn's multiple comparison test (*, P < 0.05; ns, P > 0.05). **(C)** Gating strategy used to sort lymph nodes MHCII+ CD207− FSA high dermal DCs (dDCs), MHCII+ CD207+ CD103− LCs (LCs) and MHCII+ CD207+ CD103+ cDC1 (CD103+). Red dots in the top panel depict the backgating of CD207+ LCs/cDC1. **(D)** *Atg5* mRNA expression in LCs, cDC1, and CD207− dDCs sorted from pooled lymph node cell suspensions of at least three control (*Atg5^WT^* and *Atg5^WT/Δ^*) or *Atg5^ΔCd207^* mice. Fold changes were calculated relative to mRNA expression in cells of *Atg5^WT^* control mice. ND, not detectable. **(E)** Representative dot plots for the identification of CD207+ CD11b+ LCs, CD207+ CD11b− cDC1, CD207− CD11b+ cDC2/macrophages and CD207− CD11b− (DN, double-negative) DCs among live CD45+ lineage- CD11c+ MHCII+ skin DCs in whole skin cell suspensions from *Atg5^WT^* and *Atg5^ΔCd207^* mice (lineage markers: B220, NK1.1, Ly6G and CD3). **(F)** Percentages of LCs and cDC1 among skin DCs. Data are pooled from at least three independent experiments, with each point corresponding to one individual mouse. Statistical analysis: Kruskal–Wallis one-way ANOVA followed by Dunn's multiple comparison test (*, P < 0.05; ns, P > 0.05).

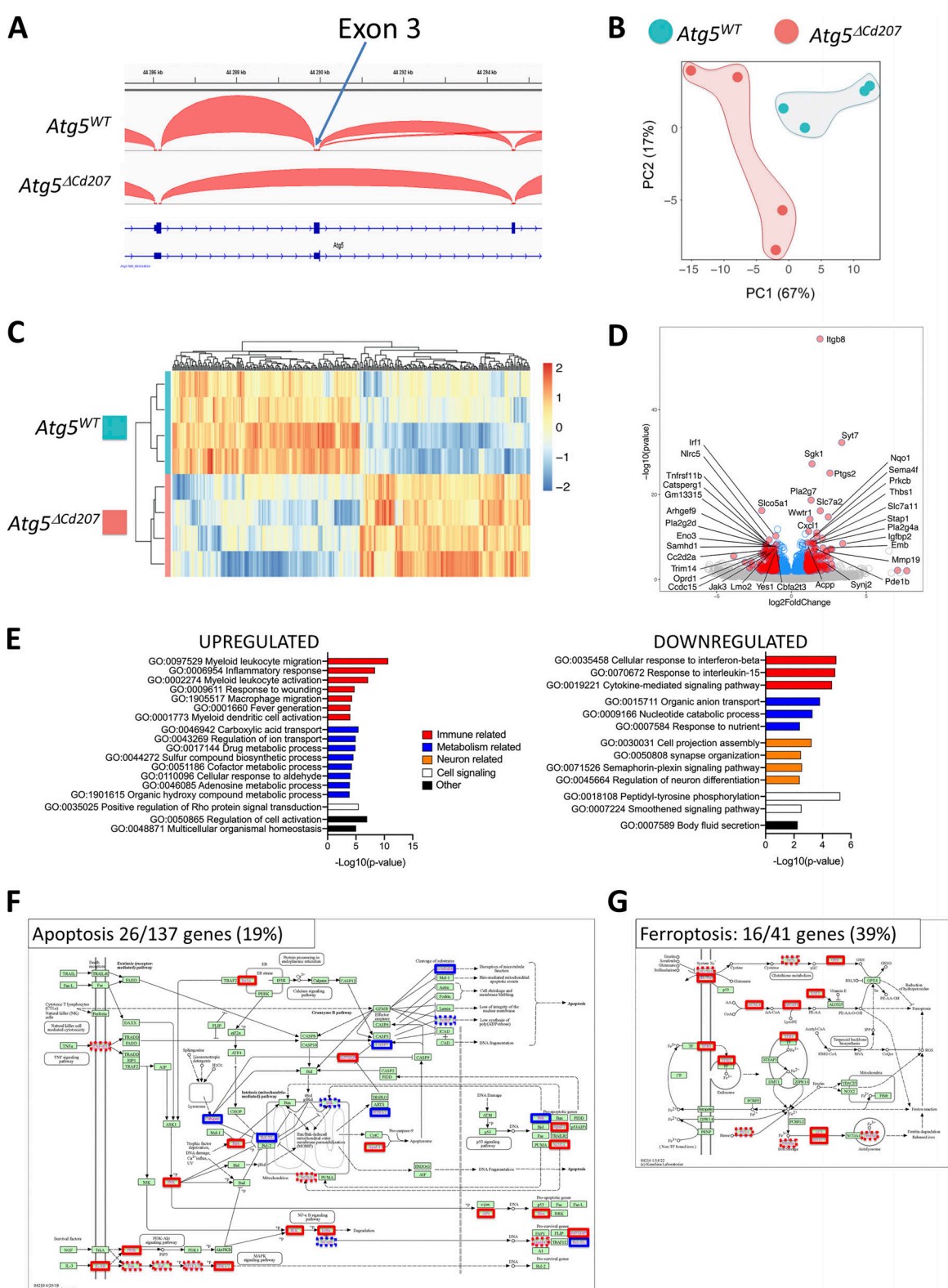

Figure S3. **Lack of autophagy alters the transcriptome of Langerhans cells. (A)** Visualization of the exon 3 region of *Atg5* gene from RNA-seq of sorted LCs of indicated mouse genotype using integrative Genomic Viewer tool. **(B)** Principal component analysis of RNA-seq transcriptome analysis from sorted LCs of *Atg5^{WT}* and A*tg5^{ΔCd207}* mice. **(C)** Heatmap showing the differentially expressed genes (FDR < 0.1, Absolute Log$_2$ Fold Change value >1, P value <0.05) between LCs of indicated mouse genotypes. **(D)** Volcano plot showing the differential expression of genes between LCs of indicated mouse genotypes. Gene names refer to the top 20 up and downregulated genes, based on the following combinations of P value and fold-change criteria: blue dots: P value <0.05 with no cutoff on Absolute Log$_2$ Fold Change; red dots: P value <0.05 and Absolute Log$_2$ Fold Change value >1. **(E)** Metascape pathway analysis of genes significantly upregulated or downregulated in *Atg5^{ΔCd207}* LCs. **(F and G)** Differentially expressed genes related to (F) apoptosis and (G) ferroptosis KEGG pathways. Blue boxes: downregulated in *Atg5^{ΔCd207}* LCs, red boxes: upregulated in *Atg5^{ΔCd207}* LCs; dashed boxes: FDR > 0.1.

Video 1.   **Autophagosome staining of *Atg5^WT* Langerhans cells.** Representative immunofluorescent staining of MAP1LC3B (LC3) on LCs obtained by in vitro emigration from epidermal sheets of *Atg5^WT* mice. LC3: green; CD207: red; DAPI: blue.

Video 2.   **Autophagosome staining of *Atg5^ΔCd207* Langerhans cells.** Representative immunofluorescent staining of MAP1LC3B (LC3) on LCs obtained by in vitro emigration from epidermal sheets of *Atg5^ΔCd207* mice. LC3: green; CD207: red; DAPI: blue.

Video 3.   **Endoplasmic reticulum staining of *Atg5^WT* Langerhans cells.** Representative immunofluorescent staining of the endoplasmic reticulum using the ER-tracker dye on epidermal LCs of 3-wk-old *Atg5^WT* mice. CD207: green; ER-tracker: red; DAPI: blue. Scale bar: 10 µm.

Video 4.   **Endoplasmic reticulum staining of *Atg5^ΔCd207* Langerhans cells.** Representative immunofluorescent staining of the endoplasmic reticulum using the ER-tracker dye on epidermal LCs of 3-wk-old *Atg5^ΔCd207* mice. CD207: green; ER-tracker: red; DAPI: blue. Scale bar: 10 µm.

Video 5.   **Lipid droplets of *Atg5^WT* Langerhans cells.** Representative immunofluorescent staining of neutral lipids using the Bodipy dye on epidermal LCs of 3-wk-old *Atg5^WT* mice. CD207: green; Bodipy: red; DAPI: blue. Scale bar: 10 µm.

Video 6.   **Lipid droplets of *Atg5^ΔCd207* Langerhans cells.** Representative immunofluorescent staining of neutral lipids using the Bodipy dye on epidermal LCs of 3-wk-old *Atg5^ΔCd207* mice. CD207: green; Bodipy: red; DAPI: blue. Scale bar: 10 µm.

**Provided online are Table S1 and Table S2. Table S1 shows differentially expressed genes between Langerhans cells of *Atg5^WT* and *Atg5^ΔCd207* mice. Table S2 shows analysis of the ferroptosis and apoptosis KEGG pathways for dysregulated genes in Langerhans cells of *Atg5^WT* versus *Atg5^ΔCd207* mice.**

