## [Peer Review File · The Journal of Cell Biology]

Epidermal maintenance of Langerhans cells relies on autophagy-regulated lipid metabolism

Florent Arbogast, Raquel Sal-Carro, Wacym Boufenghour, Quentin Frenger, Delphine Bouis, Louise Filippi de la Palavesa, Jean-Daniel Fauny, Olivier Griso, Helene Puccio, Rebecca Fima, Thierry Huby, Emmanuel Gautier, Anne Molitor, Raphael Carapito, Seiamak Bahram, Nikolaus Romani, Björn Clausen, Benjamin Voisin, Christopher Mueller, Frédéric Gros, and Vincent Flacher

Corresponding Author(s): Vincent Flacher, Institut de Biologie Moléculaire et Cellulaire and Frédéric Gros, INSERM UMR_S U1109 Immunorhumatologie Moléculaire, Federation de Medecine Translationnelle de Strasbourg (FMTS) ITI Transplantex NG, Centre de Recherche en Biomedecine de Strasbourg (CRBS); Universi

Review Timeline:

Submission Date:	2024-03-27
Editorial Decision:	2024-04-16
Revision Received:	2024-09-12
Editorial Decision:	2024-10-04
Revision Received:	2024-10-22

Monitoring Editor: Ana-María Lennon-Dumenil

Scientific Editor: Andrea Marat

Transaction Report:

DOI: <https://doi.org/10.1083/jcb.202403178>

Revision 0

Review #1

1. Evidence, reproducibility and clarity:

Evidence, reproducibility and clarity (Required)

The aim of this study was to investigate the role of autophagy in epidermal Langerhans cells (LC) by studying the phenotype of LC-specific Atg5 KO mouse that they had generated. The paper is clearly written and the authors have systematically and comprehensively assessed LCs in these mice, demonstrating that autophagy is a survival process for LCs, and that in the absence of this capability, embryo-derived LCs are lost from the skin over time.

I have no major comments about the research. Generally the experiments appear well-performed and very well-presented in the figures. These types of metabolic experiments are not in my area of expertise so I cannot judge whether they have been performed using expected controls etc.

My major comments on refer to the focus of the text which I feel somewhat obscures the impact of the paper:

1. It is not clear to me why the authors are so dogmatic about LCs being a subset of DCs. As they state in their introduction, LCs are derived from yolk sac and fetal monocyte precursors and I am not aware of any evidence that pre-DC can become LCs. Since the origin of LCs is not relevant to the findings I find that the focus of the paper on LCs as DCs distracting. Moreover, the findings that lipid metabolism is required for LC survival fits well with the tissue-resident macrophage field, in particular the recent work from the Sancho lab showing that LCs cannot survive if they don't metabolise fats correctly. To me that paper would be more relevant and interesting if presented within this context.
2. Following point 1. the authors state that autophagy has only been assessed in cDCs. I am not sure what this is being compared to - moDC? macrophages? To me the data is interesting in its own right without the need to imply that LC are related to cDC.
3. Zhang et al recent published a study in JCI Insight in which they show that LC have autophagy defects in psoriatic skin. While this paper is referenced in the current manuscript, the clear significance of the previous study in giving pathological meaning to the Atg5 KO data is missing and should be given more prominence. It is notable that Figure 4 concurs with the data published by Zhang et al showing increased lipid storage (Bodipy) and Perilipin-2 staining as a result of dysregulated autophagy.
4. The nerve data in figure 8 does not add to the manuscript and is circumstantial. It has been shown that loss of LCs leads to retraction of epidermal nerves and which is what has been repeated here - there is no mechanistic link between autophagy and neuronal interactions. I would suggest removing this figure.

****Minor points:****

1. For Figure 1, the data is from LC that have migrated out of epidermal sheets. Have the authors considered whether differences in LC3 occur upon activation/migration? Is it possible to stain for LC3 in situ in epidermal sheets?
2. There is frequent use of the expression "as expected". This makes it seem as though the findings have been shown before and I would suggest removing it.
3. The subsection titles do not match the figure titles.
4. Some figure legends need more detail e.g. for figure 4 which cells were sorted and how did they decide which genes to look at in their heatmaps, what age were the mice in different figures?
5. The headings for figures 7 and 8 are very generic - what do the authors mean by "cutaneous homeostasis"? If LCs are dying then it is the loss of LC that impacts skin function not autophagy.

2. Significance:

Significance (Required)

I believe this is an important paper that adds to our growing understanding of the importance of lipid metabolism for LC survival in the epidermis. While autophagy is not my area of expertise, and the paper is restricted to the analysis of the Atg5 KO mouse, the findings are interesting and timely and I would expect them to be of interest to the field. However, my feeling is that the dogmatic style of the paper suggesting LCs are conventional DCs could dampen interest.

3. How much time do you estimate the authors will need to complete the suggested revisions:

Estimated time to Complete Revisions (Required)

(Decision Recommendation)

Less than 1 month

4. Review Commons values the work of reviewers and encourages them to get credit for their work. Select 'Yes' below to register your reviewing activity at Web of Science Reviewer Recognition Service (formerly Publons); note that the content of your review will not be visible on Web of Science.

Yes

Review #2

1. Evidence, reproducibility and clarity:

Evidence, reproducibility and clarity (Required)

The manuscript is well written and clear, and the claims made are generally substantiated by the data, with a few exceptions.

1. In figure 5f-g, the maximal OCR and SRC for Atg5 KO LCs has an extremely wide distribution, with one outlier point having over 5x the SRC of the lower. Plotting the mean in 5f without showing the distribution is very misleading, and using a split scale in 5g creates the same issue. Including BMDCs in statistical analysis is not appropriate given their obvious differences and not suitable other than as a simple positive control. This experiment needs to be repeated with more samples, or removed.
2. In figure 8b, an additional neuronal marker should be used, e.g. GAP43.

****Minor points:****

1. In figure 1, A and B are not mentioned in the correct order in the text
2. In figure 8c, the n number used for statistical testing is not described.
3. In figure 8, the legend and figure titles do not match
4. In figure 8e, the linear regression p values need to be shown, but in general this panel is quite visually confusing and of uncertain value.
5. In the discussion, the authors claim the Akt PI3K pathway is affected by loss of autophagy, but this is only evidenced by a few genes - if they wish to make this claim, they should test this by measurement at the protein level (e.g. Akt phosphorylation).

2. Significance:

Significance (Required)

In this manuscript, Arbogast and colleagues examine the role of autophagy in the homeostasis of Langerhan's cells, resident DCs of the skin. Although autophagy has been identified as relevant in LC function, its more fundamental role in LC biology has not been explored and this area of investigation is therefore quite novel.

3. How much time do you estimate the authors will need to complete the suggested revisions:

Estimated time to Complete Revisions (Required)

(Decision Recommendation)

Between 1 and 3 months

4. Review Commons values the work of reviewers and encourages them to get credit for their work. Select 'Yes' below to register your reviewing activity at Web of Science Reviewer Recognition Service (formerly Publons); note that the content of your review will not be visible on Web of Science.

Yes

Review #3

1. Evidence, reproducibility and clarity:

Evidence, reproducibility and clarity (Required)

In the manuscript entitled "Epidermal maintenance of Langerhans cells relies on autophagy-regulated lipid metabolism", Arbogast and colleagues identified a crucial role of autophagy in regulating the survival of epidermal Langerhans cells (LCs). In particular, adopting a loss-of-functional approach based on mice specifically ablated in the autophagy essential gene, ATG5, in cells expressing CD207 the authors accumulated solid evidence on the fundamental role of autophagy in maintaining a proper lipid homeostasis, which appears fundamental in determining LCs viability and their maintenance in the epidermis.

The authors also nicely examined the transcriptional effects of the impaired autophagy in LCs and investigated the molecular and cellular alteration associated with this impairment.

I do not have any major concerns with the paper, as it is written well and adequately describes what was done using the proper methodological and statistical approaches.

****Minor comments:****

1. I would suggest including the Nrf2/Keap1 pathway in the conclusion section. This pathway plays a major role in the response to oxidative and electrophilic stress, which can activate Nrf2 in different skin innate immune cells including epidermal Langerhans cells. In addition, a deep interaction between Nrf2 and many redox sensitive inflammatory pathways exists and Nrf2 activity depends on autophagy through the autophagy-mediated degradation of Keap1 (see as an example the review from Ulsov and colleagues, <https://doi.org/10.1016/j.lfs.2021.120111>). To the best of my knowledge, while the importance of Nrf2 in oxidative damage and inflammation has been explored in skin DC, none or very limited information is available in LCs.

Of note, both heme oxygenase-1 (HO-1) and the autophagy receptor Sqstm1 (also known as p62) are well-characterized Nrf2 target genes, and the data presented from the authors indicate that both HO-1 and p62 transcripts are significantly up-regulated in Atg5 deltaCd207 LCs (Supplementary Figure S4g and Suppl. Table 2).

2. Supplementary Tables lack their name in the title and legend/information in the main text is absent.

2. Significance:

Significance (Required)

So far, very limited information is available on the role of autophagy in LCs, as well as on the consequences of constitutive autophagy impairment for LCs in vivo. Overall, the data presented in the present manuscript provide solid information on the molecular and cellular signaling pathways affected by a genetic modulation of LC autophagy, which would be of great interest for the skin biology community, but also for researchers in the autophagy field. Moreover, the study introduces and validates a mouse model (Atg5 deltaCd207) that will help future investigations of the neuroimmune axis in skin inflammation and immunity, and it also provides an in vivo/ex vivo tool for unraveling the connections between autophagy, Nrf2-Keap1 signaling and inflammation.

3. How much time do you estimate the authors will need to complete the suggested revisions:

Estimated time to Complete Revisions (Required)

(Decision Recommendation)

Less than 1 month

Yes

Revision Plan

Manuscript number: RC-2023-02276

Corresponding author(s): Vincent Flacher, Frédéric Gros

[The “revision plan” should delineate the revisions that authors intend to carry out in response to the points raised by the referees. It also provides the authors with the opportunity to explain their view of the paper and of the referee reports.]

The document is important for the editors of affiliate journals when they make a first decision on the transferred manuscript. It will also be useful to readers of the reprint and help them to obtain a balanced view of the paper.

*If you wish to submit a full revision, please use our "Full Revision" template. **It is important to use the appropriate template to clearly inform the editors of your intentions.**]*

1. General Statements [optional]

Dear Madam, Dear Sir,

Together with co-corresponding author Dr. Frédéric Gros, I submit here the preliminary revision and revision plan for our manuscript entitled “Epidermal maintenance of Langerhans cells relies on autophagy-regulated lipid metabolism” by F. Arbogast, R. Sal-Carro et al. We sincerely apologize for the slight delay of resubmission. We would like to thank the reviewers for their time and their suggestions. We have integrated most of their comments, as detailed in the next sections and highlighted in the text. We regret that some of the additional experiments that were suggested are difficult to perform in due time and would therefore significantly delay the publication of our results. A complete description of the modifications that are planned or already performed is provided below. We believe that these changes have substantially improved the quality and scope of our manuscript.

*Best regards,
Dr. Vincent Flacher, Ph.D.*

2. Description of the planned revisions

Insert here a point-by-point reply that explains what revisions, additional experimentations and analyses are planned to address the points raised by the referees.

Reviewer #1

The aim of this study was to investigate the role of autophagy in epidermal Langerhans cells (LC) by studying the phenotype of LC-specific Atg5 KO mouse that they had generated. The paper is clearly written and the authors have systematically and comprehensively assessed LCs in these mice, demonstrating that autophagy is a survival process for LCs, and that in the absence of this capability, embryo-derived LCs are lost from the skin over time.

I have no major comments about the research. Generally the experiments appear well-performed and very well-presented in the figures. These types of metabolic experiments are not in my area of expertise so I cannot judge whether they have been performed using expected controls etc.

Minor points:

1. For Figure 1, the data is from LC that have migrated out of epidermal sheets. Have the authors considered whether differences in LC3 occur upon activation/migration? Is it possible to stain for LC3 in situ in epidermal sheets?

*It is difficult to conclude about the autophagy flux with in situ stainings. To address the Reviewer's request, we will complement the findings reported in **Figure 1** by performing a comparison of LC3B and autophagy flux in fresh vs crawl-out LCs by flow cytometry.*

Reviewer #2

The manuscript is well written and clear, and the claims made are generally substantiated by the data, with a few exceptions.

Minor points:

5. In the discussion, the authors claim the Akt PI3K pathway is affected by loss of autophagy, but this is only evidenced by a few genes - if they wish to make this claim, they should test this by measurement at the protein level (e.g. Akt phosphorylation).

We understand the concern of the Reviewer. Western blots are extremely difficult given the low number of LCs that can be harvested from a mouse. However, we will perform flow cytometry of LCs with antibodies against the phosphorylated forms of Akt and S6 Kinase.

3. Description of the revisions that have already been incorporated in the transferred manuscript

Reviewer #1

Major points

1. It is not clear to me why the authors are so dogmatic about LCs being a subset of DCs. As they state in their introduction, LCs are derived from yolk sac and fetal monocyte precursors and I am not aware of any evidence that pre-DC can become LCs. Since the origin of LCs is not relevant to the findings I find that the focus of the paper on LCs as DCs distracting. Moreover, the findings that lipid metabolism is required for LC survival fits well with the tissue-resident macrophage field, in particular the recent work from the Sancho lab showing that LCs cannot survive if they don't metabolise fats correctly. To me that paper would be more relevant and interesting if presented within this context.

*We understand the Reviewer's concern that some sentences in the previous version of the manuscript might have created confusion amongst readers. We do agree that the origin of LCs clearly differs from that of cDCs¹: in the **Introduction**, we state that "Langerhans cells (LCs) are resident antigen-presenting cells (APCs) of the epidermis", rather than identifying them as cDCs. In addition, we have now removed potentially controversial statements that could suggest that LCs represent a subset of conventional DCs. For instance, in the **Abstract**, we have removed the ambiguous sentence "LCs, a proliferating DC subset with extended lifespan".*

*As opposed to macrophages, both LCs and dermal cDCs undergo maturation and migrate from the skin towards draining lymph nodes. This shared function was acknowledged because we addressed the possibility that intense emigration might explain the depletion of LCs from the epidermis. Still, we are fully aware that this dogma has been challenged lately⁵, leading many recent publications to mostly consider LCs for their resident macrophage features^{2,3}. We **discuss** Dr. Sancho's results, which indeed support our conclusion on the importance of fatty acid metabolism in LCs⁴, although the involvement of lipophagy was not within the scope of their research.*

2. Following point 1. the authors state that autophagy has only been assessed in cDCs. I am not sure what this is being compared to - moDC? macrophages? To me the data is interesting in its own right without the need to imply that LC are related to cDC.

*The comment reported here ("autophagy has only been assessed in cDCs", in the **Introduction**) relates to publications addressing autophagy in DCs or macrophages in vivo. This body of work does not give information about the consequences of autophagy impairment on LCs, which is one of the reasons that prompted us to interest in them. We rephrased this paragraph to focus on the aim to study LCs, without strict comparison with cDCs.*

3. Zhang et al recent published a study in JCI Insight in which they show that LC have autophagy defects in psoriatic skin. While this paper is referenced in the current manuscript, the clear significance of the previous study in giving pathological meaning to the Atg5 KO data is missing and should be given more prominence. It is notable that Figure 4 concurs with the data

Revision Plan

published by Zhang et al showing increased lipid storage (Bodipy) and Perilipin-2 staining as a result of dysregulated autophagy.

*We are aware of the interesting study published by Zhang et al., which investigates a model of imiquimod-driven cutaneous inflammation⁷. As pointed out by the Reviewer, these findings are in line with our results. In the revised **Discussion** of the manuscript, we have further highlighted this intriguing similarity and the associated autophagy impairment, which suggests a potential therapeutic intervention strategy and should thus undergo further investigations.*

Minor points:

2. There is frequent use of the expression "as expected". This makes it seem as though the findings have been shown before and I would suggest removing it.

We have corrected this in the manuscript.

3. The subsection titles do not match the figure titles.

As requested, we have rephrased some of the subsection and figure titles for better understanding by the reader.

4. Some figure legends need more detail e.g. for figure 4 which cells were sorted and how did they decide which genes to look at in their heatmaps, what age were the mice in different figures?

Heatmaps correspond to the differentially expressed genes of Gene Ontology or KEGG pathways relevant to metabolism, immune function, neuronal regulation... Reference numbers for each of the considered pathways are now specified in the figure legends. The age of the mice is now detailed in each figure legend. Specifically, for RNA sequencing, we used LCs from 3-week-old mice.

5. The headings for figures 7 and 8 are very generic - what do the authors mean by "cutaneous homeostasis"? If LCs are dying then it is the loss of LC that impacts skin function not autophagy.

We apologize for this mistake with figure headings, we have corrected this in the manuscript.

Reviewer #2:

Major point:

1. In figure 5f-g, the maximal OCR and SRC for Atg5 KO LCs has an extremely wide distribution, with one outlier point having over 5x the SRC of the lower. Plotting the mean in 5f without showing the distribution is very misleading, and using a split scale in 5g creates the same issue. Including BMDCs in statistical analysis is not appropriate given their obvious differences and not suitable other than as a simple positive control. [...]

*We appreciate the reviewer's careful reading of the data. The graphs in **Figure 5f** do not depict a mean but one representative experiment, as specified in the legend. To clarify this and illustrate the consistency of our results, we provide below all measurements performed with Atg5^{WT} LCs (right panels, green graphs), Atg5^{ΔCa207} LCs (right panels, red graphs) and BMDCs (left panel, blue graphs). This set of data may be included as Supplementary data upon request.*

To address the Reviewer's concerns about **Figure 5g**, we have changed the scale and verified by the Grubb's test that the highest measurement for $Atg5^{\Delta Cd207}$ LCs is not an outlier, although the total number of data points limits the power of this analysis. Finally, we chose to include the data obtained with GM-CSF-generated BMDCs because of their well-described metabolism, which relies mostly on beta-oxidation⁹. Consistent with this, Seahorse data showed no significant difference between the data obtained for $Atg5^{WT}$ LCs and BMDCs.

Minor points:

1. In figure 1, A and B are not mentioned in the correct order in the text

*We did not observe the discrepancy mentioned here, from our understanding the **Figures 1a and 1b** were mentioned in the correct order. To clarify this in the text, we have replaced the potentially confusing "upper panels" and "lower panels" by the genotype of the mice considered.*

2. In figure 8c, the n number used for statistical testing is not described.

*The n number is now clearly stated in the legend of the corresponding **Figure S6** (see below).*

3. In figure 8, the legend and figure titles do not match

We apologize for this mistake with figure headings, we have corrected it in the manuscript.

4. In figure 8e, the linear regression p values need to be shown, but in general this panel is quite visually confusing and of uncertain value.

*In **Figure 8e (now Figure S6e)**, we performed Pearson's test to assess whether there was a correlation between LC numbers and the density epidermal nerve endings. As requested; R^2 and p values are now indicated below the graph plots. Moreover, to further improve clarity, we chose to depict separately the results obtained with the three different genotypes.*

*We conclude that only $Atg5^{WT}$ mice display a clear correlation between the densities of LCs and epidermal nerves, but this was not the case for $Atg5^{\Delta Cd207}$ mice. We believe that this analysis, together with the identified gene candidates for neuronal interaction of DCs, could be of value to some readers. Yet, we agree with the Reviewer that it does not fall into the scope of the main question. Consequently, we moved these results to Supplementary material (**Figure S6**).*

Reviewer #3

In the manuscript entitled "Epidermal maintenance of Langerhans cells relies on autophagy-regulated lipid metabolism", Arbogast and colleagues identified a crucial role of autophagy in regulating the survival of epidermal Langerhans cells (LCs). In particular, adopting a loss-of-functional approach based on mice specifically ablated in the autophagy essential gene, ATG5, in cells expressing CD207 the authors accumulated solid evidence on the fundamental role of autophagy in maintaining a proper lipid homeostasis, which appears fundamental in determining LCs viability and their maintenance in the epidermis.

The authors also nicely examined the transcriptional effects of the impaired autophagy in LCs and investigated the molecular and cellular alteration associated with this impairment.

I do not have any major concerns with the paper, as it is written well and adequately describes what was done using the proper methodological and statistical approaches.

Minor comments:

1) I would suggest including the Nrf2/Keap1 pathway in the conclusion section. This pathway plays a major role in the response to oxidative and electrophilic stress, which can activate Nrf2 in different skin innate immune cells including epidermal Langerhans cells. In addition, a deep interaction between Nrf2 and many redox sensitive inflammatory pathways exists and Nrf2 activity depends on autophagy through the autophagy-mediated degradation of Keap1 (see as an example the review from Ulsov and colleagues, <https://doi.org/10.1016/j.lfs.2021.120111>). To the best of my knowledge, while the importance of Nrf2 in oxidative damage and inflammation has been explored in skin DC, none or very limited information is available in LCs.

Of note, both heme oxygenase-1 (HO-1) and the autophagy receptor Sqstm1 (also known as p62) are well-characterized Nrf2 target genes, and the data presented from the authors indicate that both HO-1 and p62 transcripts are significantly up-regulated in Atg5 deltaCd207 LCs (Supplementary Figure S4g and Suppl. Table 2).

*We thank the Reviewer for this helpful insight into this pathway. This has been incorporated in the **Discussion** of the updated manuscript.*

2) Supplementary Tables lack their name in the title and legend/information in the main text is absent.

This lack of information has been corrected in the updated manuscript.

*Finally, for the complete information of the Reviewers, we wish to point out that quantification of intracellular ferrous iron by the FerroOrange assay in LCs is now included in **Figure 6C**. These experiments were not yet completed at the time of submission, which is why they were not part of the initial manuscript. Yet, they clearly support our conclusion that ferroptosis is a major pathway implicated in the depletion of autophagy-deficient LCs.*

4. Description of analyses that authors prefer not to carry out

Reviewer #1

Major point

4. The nerve data in figure 8 does not add to the manuscript and is circumstantial. It has been shown that loss of LCs leads to retraction of epidermal nerves and which is what has been repeated here - there is no mechanistic link between autophagy and neuronal interactions. I would suggest removing this figure.

We chose to include this because the previous publications relying on complete LC depletion^{2,8} did not give any mechanistic insights on the reason why the lack of LCs affects neuronal extensions. In the context of autophagy deletion, RNA sequencing data revealed a list of neuronal interaction molecules which are down regulated in LCs early upon autophagy deletion. We believe that the community could find a great interest in investigating the molecules reported here, although this was beyond the scope of our present research. Since our observations are not conclusive and do not show a causal effect, we have chosen to move Figure 8 into the supplementary data (Figure S6). We hope that the Reviewer and the Editor can agree on keeping this figure.

Reviewer #2

Major points

1. In figure 5f-g [...] This experiment needs to be repeated with more samples, or removed.

Considering the request for additional data points, we would like to draw attention on the following issues related to this assay. First, it is difficult to yield sufficient LC numbers from the skin of 3-week-old mice to perform Seahorse assays, especially with $Atg5^{\Delta Cd207}$ mice. To reach the required minimum of 200 000 LCs and perform duplicates or triplicates, sorting of LCs pooled from up to 10 mice is required, which greatly limits the number of data points and likely creates a relatively wide distribution. Still, significant statistical comparison was possible by two-way ANOVA followed by Tukey's multiple comparison test. Second, we would also like to inform the reviewer that, since the relocation of Dr. Puccio's team into another lab, we no longer have access to the Seahorse used for these experiments. Although an alternative facility may be found, it will most likely lead to a delay in delivering additional data points and may introduce technical inconsistencies.

2. In figure 8b, an additional neuronal marker should be used, e.g. GAP43.

We chose β 3-tubulin as a well-known pan-neuronal marker for these analyses. Although we would be glad to satisfy the reviewer, we need to point out that we investigated only aging mice (up to 6 months), based on previous literature. Indeed, Zhang et al. have reported that a significant decrease of epidermal nerve endings requires LCs to be completely absent for 30 days (Langerin-DTR model)². In our model, we have tested neuronal network stainings at different ages following the nearly complete depletion of autophagy-deficient LCs. Nevertheless, we could not observe a significant impairment of the neuronal network before at least 3 months of age, possibly because $Atg5^{\Delta Cd207}$ mice retain enough LCs to support nerve growth. Since

Revision Plan

most mice in our experiments have been analyzed at 3 weeks of age, we do not currently hold a sufficient stock of old mice. Consequently, performing the requested additional stainings would postpone the publication of our results by several months. However, considering this suggestion and that of Reviewer #1, we have chosen to present this figure as supplementary information (Figure S6), because we provide it merely as an important hint for future studies and does not constitute a significant part of our conclusions on the investigated mouse model.

Bibliography

1. Kaplan, D. H. Ontogeny and function of murine epidermal Langerhans cells. *Nat. Immunol.* **18**, 1068–1075 (2017).
2. Zhang, S. *et al.* Nonpeptidergic neurons suppress mast cells via glutamate to maintain skin homeostasis. *Cell* **184**, 2151-2166.e16 (2021).
3. Sheng, J. *et al.* Fate mapping analysis reveals a novel murine dermal migratory Langerhans-like cell population. *Elife* **10**, e65412 (2021).
4. Wculek, S. K. *et al.* Oxidative phosphorylation selectively orchestrates tissue macrophage homeostasis. *Immunity* **56**, 516-530.e9 (2023).
5. Doebel, T., Voisin, B. & Nagao, K. Langerhans Cells - The Macrophage in Dendritic Cell Clothing. *Trends Immunol* **38**, 817–828 (2017).
6. Lee, H. K. *et al.* In vivo requirement for Atg5 in antigen presentation by dendritic cells. *Immunity*. **32**, 227–239 (2010).
7. Zhang, X. *et al.* Abnormal lipid metabolism in epidermal Langerhans cells mediates psoriasis-like dermatitis. *JCI Insight* **7**, e150223 (2022).
8. Doss, A. L. N. & Smith, P. G. Langerhans cells regulate cutaneous innervation density and mechanical sensitivity in mouse footpad. *Neurosci Lett* **578**, 55–60 (2014).
9. Pearce, E. J. & Everts, B. Dendritic cell metabolism. *Nat Rev Immunol* **15**, 18–29 (2015).

April 16, 2024

Re: JCB manuscript #202403178T

Dr. Vincent Flacher

Laboratory CNRS UPR3572 Immunology, Immunopathology and Therapeutic Chemistry (I2CT), Strasbourg Drug Discovery and Development Institute (IMS), Institut de Biologie Moléculaire et Cellulaire
Institut de Biologie Moléculaire et Cellulaire
2 Allée Konrad Roentgen
Strasbourg 67084
France

Dear Dr. Flacher,

Thank you for submitting your manuscript entitled "Epidermal maintenance of Langerhans cells relies on autophagy-regulated lipid metabolism" to JCB from Review Commons. We have now assessed your study, the reviewers comments, and your revision plan, and agree that a revised study as outlined is suitable for further consideration at JCB. Therefore, we invite you to submit your revised manuscript to be assessed by the original reviewers, or if they are unavailable suitable replacements.

GENERAL GUIDELINES:

Text limits: Character count for an Transfer is < 40,000, not including spaces. Count includes title page, abstract, introduction, results, discussion, and acknowledgments. Count does not include materials and methods, figure legends, references, tables, or supplemental legends.

Figures: Transfers may have up to 10 main text figures. Figures must be prepared according to the policies outlined in our Instructions to Authors, under Data Presentation, <https://jcb.rupress.org/site/misc/ifora.xhtml>. All figures in accepted manuscripts will be screened prior to publication.

Supplemental information: There are strict limits on the allowable amount of supplemental data. Transfers may have up to 5 supplemental figures. Up to 10 supplemental videos or flash animations are allowed. A summary of all supplemental material should appear at the end of the Materials and methods section.

Please note that JCB now requires authors to submit Source Data used to generate figures containing gels and Western blots with all revised manuscripts. This Source Data consists of fully uncropped and unprocessed images for each gel/blot displayed in the main and supplemental figures. Since your paper includes cropped gel and/or blot images, please be sure to provide one Source Data file for each figure that contains gels and/or blots along with your revised manuscript files. File names for Source Data figures should be alphanumeric without any spaces or special characters (i.e., SourceDataF#, where F# refers to the associated main figure number or SourceDataFS# for those associated with Supplementary figures). The lanes of the gels/blots should be labeled as they are in the associated figure, the place where cropping was applied should be marked (with a box), and molecular weight/size standards should be labeled wherever possible.

The typical timeframe for revisions is three to four months. While most universities and institutes have reopened labs and allowed researchers to begin working at nearly pre-pandemic levels, we at JCB realize that the lingering effects of the COVID-19 pandemic may still be impacting some aspects of your work, including the acquisition of equipment and reagents. Therefore, if you anticipate any difficulties in meeting this aforementioned revision time limit, please contact us and we can work with you to find an appropriate time frame for resubmission. Please note that papers are generally considered through only one revision cycle, so any revised manuscript will likely be either accepted or rejected.

Thank you for this interesting contribution to Journal of Cell Biology. You can contact us at the journal office with any questions at cellbio@rockefeller.edu.

Sincerely,

Ana-María Lennon-Dumenil, PhD
Monitoring Editor

Andrea L. Marat, PhD
Senior Scientific Editor

Journal of Cell Biology

Point-by-point reply to the Reviewers

Reviewer #1

The aim of this study was to investigate the role of autophagy in epidermal Langerhans cells (LC) by studying the phenotype of LC-specific Atg5 KO mouse that they had generated. The paper is clearly written and the authors have systematically and comprehensively assessed LCs in these mice, demonstrating that autophagy is a survival process for LCs, and that in the absence of this capability, embryo-derived LCs are lost from the skin over time.

I have no major comments about the research. Generally the experiments appear well-performed and very well-presented in the figures. These types of metabolic experiments are not in my area of expertise so I cannot judge whether they have been performed using expected controls etc.

Major points

1. It is not clear to me why the authors are so dogmatic about LCs being a subset of DCs. As they state in their introduction, LCs are derived from yolk sac and fetal monocyte precursors and I am not aware of any evidence that pre-DC can become LCs. Since the origin of LCs is not relevant to the findings I find that the focus of the paper on LCs as DCs distracting. Moreover, the findings that lipid metabolism is required for LC survival fits well with the tissue-resident macrophage field, in particular the recent work from the Sancho lab showing that LCs cannot survive if they don't metabolise fats correctly. To me that paper would be more relevant and interesting if presented within this context.

*We understand the Reviewer's concern that some sentences in the previous version of the manuscript might have created confusion amongst readers. We do agree that the origin of LCs clearly differs from that of cDCs¹: in the **Introduction**, we state that "Langerhans cells (LCs) are resident antigen-presenting cells (APCs) of the epidermis", rather than identifying them as cDCs. In addition, we have now removed potentially controversial statements that could suggest that LCs represent a subset of conventional DCs. For instance, in the **Abstract**, we have removed the ambiguous sentence "LCs, a proliferating DC subset with extended lifespan".*

*As opposed to macrophages, both LCs and dermal cDCs undergo maturation and migrate from the skin towards draining lymph nodes. This shared function was acknowledged because we addressed the possibility that intense emigration might explain the depletion of LCs from the epidermis. Still, we are fully aware that this dogma has been challenged lately⁵, leading many recent publications to mostly consider LCs for their resident macrophage features^{2,3}. We **discuss** Dr. Sancho's results, which indeed support our conclusion on the importance of fatty acid metabolism in LCs⁴, although the involvement of lipophagy was not within the scope of their research.*

2. Following point 1. the authors state that autophagy has only been assessed in cDCs. I am not sure what this is being compared to - moDC? macrophages? To me the data is interesting in its own right without the need to imply that LC are related to cDC.

*The comment reported here by the Reviewer ("autophagy has only been assessed in cDCs", in the **Introduction**) relates to publications addressing autophagy in DCs or macrophages in vivo. This body of work does not give information about the consequences of autophagy impairment on LCs, which is one of the reasons that prompted us to interest in them. We rephrased this paragraph to focus on the aim to study LCs, without strict comparison with cDCs.*

3. Zhang et al recent published a study in JCI Insight in which they show that LC have autophagy defects in psoriatic skin. While this paper is referenced in the current manuscript, the clear significance of the previous study in giving pathological meaning to the Atg5 KO data is missing and should be given more prominence. It is notable that Figure 4 concurs with the data published by Zhang et al showing increased lipid storage (Bodipy) and Perilipin-2 staining as a result of dysregulated autophagy.

*We are aware of the interesting study published by Zhang et al. investigating a model of imiquimod-driven cutaneous inflammation, and it was cited in our manuscript⁷. As pointed out by the Reviewer, these findings are in line with our results. In the revised **Discussion** of the manuscript, we have further*

highlighted this intriguing similarity and the associated autophagy impairment, which suggests a potential therapeutic intervention strategy and should thus undergo further investigations.

4. The nerve data in figure 8 does not add to the manuscript and is circumstantial. It has been shown that loss of LCs leads to retraction of epidermal nerves and which is what has been repeated here - there is no mechanistic link between autophagy and neuronal interactions. I would suggest removing this figure.

*We chose to include this because the previous publications, relying on complete LC depletion^{2,8}, did not give any mechanistic insights on the reason why the lack of LCs affects neuronal extensions. On the other hand, in the context of autophagy deletion, our RNA sequencing data revealed a list of neuronal interaction molecules which are down regulated in LCs early upon autophagy deletion. Although our observations are not conclusive, we believe that the community could find a great interest in investigating the molecules reported here, which was beyond the scope of our present research project. We hope that the Reviewer and the Editor will agree on keeping this figure (now **Figure 9**).*

Minor points:

1. For Figure 1, the data is from LC that have migrated out of epidermal sheets. Have the authors considered whether differences in LC3 occur upon activation/migration? Is it possible to stain for LC3 in situ in epidermal sheets?

*It is difficult to conclude about the autophagy flux with in situ staining. To address the Reviewer's request, we have complemented the findings reported in **Figure 1B-C** by performing a comparison of LC3B and autophagy flux by flow cytometry in both freshly isolated (immature) and crawl-out (mature) LCs. The results did not show any significant difference of autophagy flux related to the maturation state.*

2. There is frequent use of the expression "as expected". This makes it seem as though the findings have been shown before and I would suggest removing it.

We have corrected this in the manuscript.

3. The subsection titles do not match the figure titles.

As requested, we have rephrased some of the subsection and figure titles for better understanding by the reader.

4. Some figure legends need more detail e.g. for figure 4 which cells were sorted and how did they decide which genes to look at in their heatmaps, what age were the mice in different figures?

Heatmaps correspond to the differentially expressed genes of Gene Ontology or KEGG pathways relevant to metabolism, immune function, neuronal regulation... Reference numbers for each of the considered pathways are now specified in the figure legends. The age of the mice is now detailed in each figure legend. Specifically, for RNA sequencing, we used LCs from 3-week-old mice.

5. The headings for figures 7 and 8 are very generic - what do the authors mean by "cutaneous homeostasis"? If LCs are dying then it is the loss of LC that impacts skin function not autophagy.

We apologize for this mistake with figure headings, we have corrected this in the manuscript.

Reviewer #2:

The manuscript is well written and clear, and the claims made are generally substantiated by the data, with a few exceptions.

Major point:

1. In figure 5f-g, the maximal OCR and SRC for Atg5 KO LCs has an extremely wide distribution, with one outlier point having over 5x the SRC of the lower. Plotting the mean in 5f without showing the distribution is very misleading, and using a split scale in 5g creates the same issue. Including BMDCs in statistical analysis is not appropriate given their obvious differences and not suitable other than as a simple positive control. This experiment needs to be repeated with more samples, or removed.

We appreciate the reviewer's careful reading of the data. The graphs in Figure 6F (formerly Figure 5F) do not depict a mean but one representative experiment, as specified in the legend. To clarify this and illustrate the consistency of our results, we provide below all measurements performed with Atg5^{WT} LCs (right panels, green graphs), Atg5^{ΔCd207} LCs (right panels, red graphs) and BMDCs (left panel, blue graphs). This set of data may be included as Supplementary data upon request.

To address the Reviewer's concerns about Figure 6G, we have changed the scale and verified by the Grubb's test that the highest measurement for Atg5^{ΔCd207} LCs is not an outlier. Finally, we chose to include the data obtained with GM-CSF-generated BMDCs because of their well-described metabolism, which relies mostly on beta-oxidation⁹. Consistent with this, Seahorse data showed no significant difference between the data obtained for Atg5^{WT} LCs and BMDCs.

Considering the request for additional data points, we would like to draw attention on the following issues related to this assay. First, it is difficult to yield sufficient LC numbers from the skin of 3-week-old mice to perform Seahorse assays, especially with Atg5^{ΔCd207} mice. To reach the required minimum of 200 000 LCs and perform duplicates or triplicates, sorting of LCs pooled from up to 10 mice was required, which greatly limited the number of data points and likely created a relatively wide distribution. Still, significant statistical comparison was possible by two-way ANOVA followed by Tukey's multiple comparison test. Second, we also need to inform the Reviewer that, since the relocation of coauthor Dr. Puccio into another lab, we no longer have access to the Seahorse used for these experiments. We could not find an alternative facility to perform these assays within the three months that were necessary for the revision.

2. In figure 8b, an additional neuronal marker should be used, e.g. GAP43.

We chose β3-tubulin as a well-known pan-neuronal marker for these analyses. Although we would be glad to fulfill this request, we need to point out that we investigated only aging mice (up to 6 months), based on previous literature. Indeed, Zhang et al. have reported that a significant decrease of epidermal nerve endings requires LCs to be completely absent for 30 days (Langerin-DTR model)². In our model, we have tested neuronal network staining at different ages following the nearly complete depletion of autophagy-deficient LCs. Nevertheless, we could not observe a significant impairment of the neuronal network before at least 3 months of age, possibly because Atg5^{ΔCd207} mice retain enough LCs to support nerve growth. Since most mice in our experiments have been analyzed at 3 weeks of age, we do not currently hold a sufficient stock of old mice. Consequently, performing the requested additional staining would have postponed the manuscript revision by several months, which was most likely not acceptable for the JCB. Finally, we provide the data in Figure 9 (formerly Figure 8) merely as

an important hint for future studies, as it does not significantly contribute to our conclusions on the investigated mouse model.

Minor points:

1. In figure 1, A and B are not mentioned in the correct order in the text

*We were unable to find the discrepancy mentioned here: from our understanding the **Figures 1A and 1B** were mentioned in the correct order. To clarify this in the text, we have replaced the potentially confusing “upper panels” and “lower panels” by the genotype of the mice considered.*

2. In figure 8c, the n number used for statistical testing is not described.

*The n number is now clearly stated in the legend of the corresponding **Figure 9** (see below).*

3. In figure 8, the legend and figure titles do not match

We apologize for this mistake with figure headings, we have corrected it in the manuscript.

4. In figure 8e, the linear regression p values need to be shown, but in general this panel is quite visually confusing and of uncertain value.

*In **Figure 9E** (formerly **Figure 8E**), we performed Pearson’s test to assess whether there was a correlation between LC numbers and the density epidermal nerve endings. As requested, R^2 and p values are now indicated below the graph plots. Moreover, to further improve clarity, we chose to depict separately the results obtained with the three different genotypes.*

We conclude that only $Atg5^{WT}$ mice display a clear correlation between the densities of LCs and epidermal nerves, but this was not the case for $Atg5^{ΔCd207}$ mice. We agree with the Reviewer that this analysis does not fall into the scope of our main question, yet we believe that, together with the identified gene candidates for neuronal interaction of DCs, this information could be valuable to readers interested in neuro-immune crosstalk.

5. In the discussion, the authors claim the Akt PI3K pathway is affected by loss of autophagy, but this is only evidenced by a few genes - if they wish to make this claim, they should test this by measurement at the protein level (e.g. Akt phosphorylation).

*We understand the concern of the Reviewer. A Western Blot would be extremely difficult to achieve, given the low number of LCs that can be harvested from a mouse. However, we have performed flow cytometry of LCs with antibodies against the phosphorylated forms of Akt, 4E-BP1 and S6 (**Figure 4**) and demonstrated a constitutive activation of the PI3K/Akt pathway in autophagy-deficient LCs.*

Reviewer #3

In the manuscript entitled "Epidermal maintenance of Langerhans cells relies on autophagy-regulated lipid metabolism", Arbogast and colleagues identified a crucial role of autophagy in regulating the survival of epidermal Langerhans cells (LCs). In particular, adopting a loss-of-functional approach based on mice specifically ablated in the autophagy essential gene, ATG5, in cells expressing CD207 the authors accumulated solid evidence on the fundamental role of autophagy in maintaining a proper lipid homeostasis, which appears fundamental in determining LCs viability and their maintenance in the epidermis.

The authors also nicely examined the transcriptional effects of the impaired autophagy in LCs and investigated the molecular and cellular alteration associated with this impairment.

I do not have any major concerns with the paper, as it is written well and adequately describes what was done using the proper methodological and statistical approaches.

Minor comments:

1) I would suggest including the Nrf2/Keap1 pathway in the conclusion section. This pathway plays a major role in the response to oxidative and electrophilic stress, which can activate Nrf2 in different skin innate immune cells including epidermal Langerhans cells. In addition, a deep interaction between Nrf2 and many redox sensitive inflammatory pathways exists and Nrf2 activity depends on autophagy through the autophagy-mediated degradation of Keap1 (see as an example the review from Ulsov and colleagues, <https://doi.org/10.1016/j.lfs.2021.120111>). To the best of my knowledge, while the importance of Nrf2 in oxidative damage and inflammation has been explored in skin DC, none or very limited information is available in LCs.

Of note, both heme oxygenase-1 (HO-1) and the autophagy receptor Sqstm1 (also known as p62) are well-characterized Nrf2 target genes, and the data presented from the authors indicate that both HO-1 and p62 transcripts are significantly up-regulated in Atg5 deltaCd207 LCs (Supplementary Figure S4g and Suppl. Table 2).

We thank the Reviewer for this helpful insight into this pathway. This has been incorporated in the Discussion of the updated manuscript.

2) Supplementary Tables lack their name in the title and legend/information in the main text is absent.

This lack of information has been corrected in the updated manuscript.

Additional data on ferroptosis

*Finally, for the complete information of Reviewers, we wish to point out that quantification of intracellular ferrous iron by the FerroOrange assay in LCs is now included in **Figure 7C**. These experiments were not yet completed at the time of submission, which is why they were not part of the initial manuscript. Yet, they clearly support our conclusion that ferroptosis is a major pathway implicated in the depletion of autophagy-deficient LCs.*

Bibliography

1. Kaplan, D. H. Ontogeny and function of murine epidermal Langerhans cells. *Nat.Immunol.* **18**, 1068–1075 (2017).
2. Zhang, S. *et al.* Nonpeptidergic neurons suppress mast cells via glutamate to maintain skin homeostasis. *Cell* **184**, 2151-2166.e16 (2021).
3. Sheng, J. *et al.* Fate mapping analysis reveals a novel murine dermal migratory Langerhans-like cell population. *Elife* **10**, e65412 (2021).
4. Wculek, S. K. *et al.* Oxidative phosphorylation selectively orchestrates tissue macrophage homeostasis. *Immunity* **56**, 516-530.e9 (2023).
5. Doebel, T., Voisin, B. & Nagao, K. Langerhans Cells - The Macrophage in Dendritic Cell Clothing. *Trends Immunol* **38**, 817–828 (2017).
6. Lee, H. K. *et al.* In vivo requirement for Atg5 in antigen presentation by dendritic cells. *Immunity*. **32**, 227–239 (2010).
7. Zhang, X. *et al.* Abnormal lipid metabolism in epidermal Langerhans cells mediates psoriasis-like dermatitis. *JCI Insight* **7**, e150223 (2022).
8. Doss, A. L. N. & Smith, P. G. Langerhans cells regulate cutaneous innervation density and mechanical sensitivity in mouse footpad. *Neurosci Lett* **578**, 55–60 (2014).
9. Pearce, E. J. & Everts, B. Dendritic cell metabolism. *Nat Rev Immunol* **15**, 18–29 (2015).

October 4, 2024

RE: JCB Manuscript #202403178R

Dr. Vincent Flacher
Laboratory CNRS UPR3572 Immunology, Immunopathology and Therapeutic Chemistry (I2CT), Strasbourg Drug Discovery and Development Institute (IMS), Institut de Biologie Moléculaire et Cellulaire
Institut de Biologie Moléculaire et Cellulaire
2 Allée Konrad Roentgen
Strasbourg 67084
France

Dear Dr. Flacher,

Thank you for submitting your revised manuscript entitled "Epidermal maintenance of Langerhans cells relies on autophagy-regulated lipid metabolism". We would be happy to publish your paper in JCB pending final revisions necessary to meet our formatting guidelines (see details below). In your final text, you must also address the final reviewer comments.

A. MANUSCRIPT ORGANIZATION AND FORMATTING:

- 1) Text limits: Character count for Articles is < 40,000, not including spaces. Count includes abstract, introduction, results, discussion, and acknowledgments. Count does not include title page, figure legends, materials and methods, references, tables, or supplemental legends.
- 2) Figures limits: Articles may have up to 10 main text figures.
- 3) Figure formatting: Scale bars must be present on all microscopy images, including inset magnifications. Molecular weight or nucleic acid size markers must be included on all gel electrophoresis. Aspect ratios of images may not be altered. In order to accommodate readers with red-green color blindness, we suggest that you change all red/green color schemes.
- 4) Statistical analysis: Error bars on graphic representations of numerical data must be clearly described in the figure legend. The number of independent data points (n) represented in a graph must be indicated in the legend. Statistical methods should be explained in full in the materials and methods. For figures presenting pooled data the statistical measure should be defined in the figure legends. Please also be sure to indicate the statistical tests used in each of your experiments (either in the figure legend itself or in a separate methods section) as well as the parameters of the test (for example, if you ran a t-test, please indicate if it was one- or two-sided, etc.). Also, if you used parametric tests, please indicate if the data distribution was tested for normality (and if so, how). If not, you must state something to the effect that "Data distribution was assumed to be normal but this was not formally tested."
- 5) Abstract and title: The abstract should be no longer than 160 words and should communicate the significance of the paper for a general audience. The title should be less than 100 characters including spaces. Make the title concise but accessible to a general readership.
- 6) Materials and methods: Should be comprehensive and not simply reference a previous publication for details on how an experiment was performed. Please provide full descriptions in the text for readers who may not have access to referenced manuscripts.
- 7) All antibodies, cell lines, animals, and tools used in the manuscript should be described in full, including accession numbers for materials available in a public repository such as the Resource Identification Portal. Please be sure to provide the sequences for all of your primers/oligos and RNAi constructs in the materials and methods. You must also indicate in the methods the source, species, and catalog numbers (where appropriate) for all of your antibodies. Please also indicate the acquisition and quantification methods for immunoblotting/western blots.
- 8) Microscope image acquisition: The following information must be provided about the acquisition and processing of images:
 - a. Make and model of microscope
 - b. Type, magnification, and numerical aperture of the objective lenses

- c. Temperature
- d. Imaging medium
- e. Fluorochromes
- f. Camera make and model
- g. Acquisition software
- h. Any software used for image processing subsequent to data acquisition. Please include details and types of operations involved (e.g., type of deconvolution, 3D reconstitutions, surface or volume rendering, gamma adjustments, etc.).

10) Supplemental materials: There are strict limits on the allowable amount of supplemental data. Articles may have up to 5 supplemental figures. Please also note that tables, like figures, should be provided as individual, editable files. A summary of all supplemental material should appear at the end of the Materials and methods section.

13) ORCID IDs: ORCID IDs are unique identifiers allowing researchers to create a record of their various scholarly contributions in a single place. Please note that ORCID IDs are now *required* for all authors. At resubmission of your final files, please be sure to provide your ORCID ID and those of all co-authors.

Please note that JCB now requires authors to submit Source Data used to generate figures containing gels and Western blots with all revised manuscripts. This Source Data consists of fully uncropped and unprocessed images for each gel/blot displayed in the main and supplemental figures. File names for Source Data figures should be alphanumeric without any spaces or special characters (i.e., SourceDataF#, where F# refers to the associated main figure number or SourceDataFS# for those associated with Supplementary figures). The lanes of the gels/blots should be labeled as they are in the associated figure, the place where cropping was applied should be marked (with a box), and molecular weight/size standards should be labeled wherever possible. Source Data files will be made available to reviewers during evaluation of revised manuscripts and, if your paper is eventually published in JCB, the files will be directly linked to specific figures in the published article.

Journal of Cell Biology now requires a data availability statement for all research article submissions. These statements will be published in the article directly above the Acknowledgments. The statement should address all data underlying the research presented in the manuscript. Please visit the JCB instructions for authors for guidelines and examples of statements at (<https://rupress.org/jcb/pages/editorial-policies#data-availability-statement>).

B. FINAL FILES:

****It is JCB policy that if requested, original data images must be made available to the editors. Failure to provide original images upon request will result in unavoidable delays in publication. Please ensure that you have access to all original data images prior to final submission.****

****The license to publish form must be signed before your manuscript can be sent to production. A link to the electronic license to publish form will be sent to the corresponding author only. Please take a moment to check your funder requirements before choosing the appropriate license.****

Thank you for your attention to these final processing requirements. Please revise and format the manuscript and upload materials within 7 days. If you need an extension for whatever reason, please let us know and we can work with you to determine a suitable revision period.

Thank you for this interesting contribution, we look forward to publishing your paper in Journal of Cell Biology.

Sincerely,

Ana-Maria Lennon-Dumenil, PhD
Monitoring Editor

Andrea L. Marat, PhD
Deputy Editor

Journal of Cell Biology

Reviewer #1 (Comments to the Authors (Required)):

This is a revision of the authors study investigating the role of autophagy in epidermal Langerhans cells (LC) by analysing the phenotype of the LC-specific Atg5 KO mouse. The authors have worked hard to address the reviewers' comments and I congratulate them on an improved manuscript.

My only minor comments refer to Figure 9:

1. The title for Figure 9 does not match the heading used for that section in the text and is inaccurate. The current heading implies active depletion of LCs whilst the authors have merely quantified nerves in the epidermis of mice from which LCs are gradually lost due to Atg5 depletion. This should be changed.
2. I still believe that, whilst potentially interesting, the neuronal data is superfluous to the main message of the manuscript, which already has 8 figures. If the authors are keen to keep this figure then it would be interesting to discuss why they see a putative loss of "neuronal" genes at 3 weeks of age in Atg5 KO LCs but do not see a difference in the nerves until 6-12 months (as discussed in response to the reviewers).
3. The authors argue that the lack of correlation between LC and nerve numbers in the Atg5KO mice implies that the LCs in these mice are no longer able to interact with nerves. However, have they considered that there is a lack of correlation simply because there are not enough LCs or nerve endings at the 6-12 month time points to obtain statistically robust data? Considering also my point 2. above, I do not agree that authors can infer that functional changes in LCs cause loss of nerves from these data.

Reviewer #2 (Comments to the Authors (Required)):

I am satisfied with the authors' responses to my comments.

Reviewer #1 (Comments to the Authors (Required)):

This is a revision of the authors study investigating the role of autophagy in epidermal Langerhans cells (LC) by analysing the phenotype of the LC-specific Atg5 KO mouse. The authors have worked hard to address the reviewers' comments and I congratulate them on an improved manuscript.

We thank the Reviewer for this appreciation of our revisions.

My only minor comments refer to Figure 9:

1. The title for Figure 9 does not match the heading used for that section in the text and is inaccurate. The current heading implies active depletion of LCs whilst the authors have merely quantified nerves in the epidermis of mice from which LCs are gradually lost due to Atg5 depletion. This should be changed.

We have modified the Figure title into a statement reflecting our data and conclusions with more accuracy:

Section heading: Autophagy deficiency affects neuronal interaction genes in Langerhans cells

Figure 9 title: Depletion of autophagy-deficient Langerhans cells alters the epidermal neuronal network.

New Figure 9 title: Autophagy deficiency affects neuronal interaction genes in Langerhans cells

2. I still believe that, whilst potentially interesting, the neuronal data is superfluous to the main message of the manuscript, which already has 8 figures. If the authors are keen to keep this figure then it would be interesting to discuss why they see a putative loss of "neuronal" genes at 3 weeks of age in Atg5 KO LCs but do not see a difference in the nerves until 6-12 months (as discussed in response to the reviewers).

The JCB instructions for authors state allow for 10 main figures and 5 supplemental figures. Following the Reviewer's suggestion, we have introduced in the Discussion a statement derived from our previous reply to Major point #2 of Reviewer #2, acknowledging the limitations of our conclusions and calling for further investigations.

3. The authors argue that the lack of correlation between LC and nerve numbers in the Atg5KO mice implies that the LCs in these mice are no longer able to interact with nerves. However, have they considered that there is a lack of correlation simply because there are not enough LCs or nerve endings at the 6-12 month time points to obtain statistically robust data? Considering also my point 2. above, I do not agree that authors can infer that functional changes in LCs cause loss of nerves from these data.

Altogether, we agree with the Reviewer that transcriptomic data and a lack of correlation are not sufficient to prove that the altered expression of neuronal interactions molecules by autophagy-deficient LCs directly causes the loss of epidermal nerve endings. An unequivocal demonstration would require additional mechanistic experiments that would be of great interest but were beyond the scope of our present manuscript and our current research. Nevertheless, we cannot exclude that this piece of data may be useful for future research projects, which is why we are providing it to the community. Taking into account the concerns expressed by the Reviewer, we have taken further precautions in the Discussion to prevent overinterpretation of our data.

Reviewer #2 (Comments to the Authors (Required)):

I am satisfied with the authors' responses to my comments.

We thank the Reviewer for the helpful comments that allowed us to improve our manuscript.